# *In-silico* platform for the multifunctional design of 3D printed conductive components

Javier Crespo-Miguel [1], Sergio Lucarini [2,3,4], Sara Garzon-Hernandez [1], Angel Arias[1], Emilio Martínez-Pañeda [4,5] & Daniel Garcia-Gonzalez [1] ✉

The effective electric resistivity of conductive thermoplastics manufactured by filament extrusion methods is determined by both the material constituents and the printing parameters. The former determines the multifunctional nature of the composite, whereas the latter dictates the mesostructural characteristics such as filament adhesion and void distribution. This work provides a multi-scale computational framework to evaluate the thermo-electro-mechanical behaviour of printed conductive polymers. A full-field homogenisation model first provides the influence of material and mesostructural features (i.e., filament orientations, voids and adhesion between filaments). Then, a macroscopic continuum model elucidates the effects of thermo-electro-mechanical mixed boundary conditions. The *in-silico* multi-scale methodology is validated with extensive original multi-physical experiments and a functional application consisting of an electro-heatable printing cartridge. Overall, this work establishes the foundations to virtually break the gap between mesoscopic and macroscopic multifunctional responses in conductive components manufactured by additive manufacturing techniques.

Recent advances in additive manufacturing (AM) techniques have enabled the design of geometrically customised components with electric conductive properties. These methods commonly use conductive polymer composites (CPCs) in the form of filaments, which usually consist of conductive fillers within a thermoplastic matrix[1–6]. By adding a sufficient amount of conductive filler to the matrix, conductive pathways are created at the microscale, making effective macroscopic current flows possible[7–10]. The current flows within the material lead to an increase in the kinetic energy derived from the electron movement, resulting in Joule heating[11–16]. This interplay introduces a thermo-mechanical response that is especially relevant when using polymeric matrices with highly temperature-dependent properties[17,18]. Likewise, mechanical deformation of the material modulates the effective electrical properties by changing the relative position of particle clusters and the associated conductive paths[19–24]. This results in an interrelationship between electrical, thermal and

mechanical responses, requiring a multi-physical analysis to understand the behaviour of CPC-based conductive components[25,26]. In addition, the use of AM technologies to manufacture conductive components incorporates additional dependencies related to the printing process and its impact on mesostructural defects[27–31]. In this regard, components manufactured by Fused Filament Fabrication (FFF) exhibit a macroscopic orthotropic response due to the formation of mesoporous between adjacent filaments[32,33]. This orthotropy is determined by the printing orientation and makes the multi-physical response of printed components highly dependent on the mixed boundary conditions (i.e., thermal, electrical and mechanical)[34].

Previous research works have proposed homogenisation approaches to predict the effect of mesostructural defects, derived from the printing process, on the purely mechanical behaviour[35–39]. The mechano-electrical[40,41] and electro-thermal[42] responses of isotropic CPCs have been modelled independently. However, the use of

[1]Department of Continuum Mechanics and Structural Analysis, University Carlos III of Madrid, Avda. de la Universidad 30, Leganés, Madrid, Spain. [2]BCMaterials, Basque Center for Materials, Applications and Nanostructures, UPV/EHU Science Park, Leioa, Spain. [3]Ikerbasque, Basque Foundation for Science, Bilbao, Spain. [4]Department of Civil and Environmental Engineering, Imperial College of London, South Kensington Campus, London, UK. [5]Department of Engineering Science, University of Oxford, Oxford, UK. ✉e-mail: danigarc@ing.uc3m.es

FFF to manufacture conductive components poses the need to account for preferred directions and material heterogeneity (i.e., along the filament deposition and the adhesion between adjacent filaments). Although this problem has not been addressed yet, previous experimental work has provided insights into the multi-physical response of CPC-materials printed by FFF[43–47]. These studies show a variation in electric resistance depending on different printing parameters such as layer height[31] or printing direction[48,49]. Despite these observations, the mechanisms and links between the multiphysics response of printed conductive components and their mesostructure remain elusive. To address this need, a combination of numerical and experimental approaches is essential.

In this work, we introduce *in-silico* tools to predict the thermo-electro-mechanical interplay of 3D printed conductive components, providing a route map for the modulation of their multi-physical response during FFF printing. We experimentally identify the main mechanisms governing the orthotropic multi-physical behaviour of FFF components, and introduce them into a multi-scale computational approach that links the multi-functional responses at the meso- and macroscales. The experimental approach first studies the inter-dependencies between physics by pairs, isolating them one-to-one (i.e., mechano-electrical, electro-thermal and thermo-mechanical interplays). Then, the experiments are extended to fully-coupled tests in which all studied fields (i.e., thermal, electrical and mechanical) are considered simultaneously. Samples with three different printing orientations have been used throughout the experimental campaign, i.e., longitudinal, transverse and oblique, using a conductive 3D printing filament based on polylactic acid (PLA) and Carbon Black (CB) particles. The numerical approach comprises different modelling stages at the two scales of the problem, i.e., meso and macroscales. A full-field homogenisation framework has been used to unravel the effect of the mesostructure (derived from the printing process) on the mechano-electrical response of FFF components. At the macroscopic scale, a continuum model has been developed accounting for the mechano-electrical orthotropy present in FFF components as well as its electro-thermal interdependence, considering Joule heating and convective terms. The combination of both scales allows to design and simulate the behaviour of the 3D printed conductive component, optimising its multifunctional performance by tuning the printing patterns. The complete framework has been proven to be a valuable tool to guide the design of 3D printed conductive devices with experimental demonstrations at the functional level.

## Results

### Experimental testing platform to evaluate the multi-physical behaviour of 3D printed conductive polymers

The conductive capabilities of a CPC are given by the interrelation between conductive particles at the microscale. The presence of defects (i.e., voids or heterogeneous particle distributions) prevents from the optimal formation of conductive pathways. The FFF printing process introduces mesostructural inter-filament voids in a preferred direction due to the imperfect adhesion between deposited filaments. Therefore, the choice of the printing parameters modulates the multiphysical performance of a printed part. To study the effect of the printing direction, one of the most influencing parameters in the thermo-electro-mechanical behaviour, we manufactured rectangular samples considering three directions (see Fig. 1a): (i) longitudinal, presenting a parallel printing direction with respect to mechanical and electrical loading; (ii) transverse, presenting a perpendicular printing direction with respect to the loading; and (iii) oblique, presenting a printing direction rotated 45° with respect to the loading. To identify the phenomena that take place in the multi-physical interplay of components manufactured by FFF, we provide an extensive experimental campaign accounting for the thermal, electrical and mechanical responses. The testing platform was conceived to isolate the

couplings by pairs of physics, resulting in three independent sets of tests: (i) electro-thermal; (ii) mechano-electrical; and (iii) thermo-mechanical. In addition, a fully-coupled experiment was performed, controlling/measuring every studied variable simultaneously (i.e., temperature, electrical resistivity and displacement).

To apply an electric field to the samples, ad-hoc electrode grips were attached to the universal testing machine. The grips allowed for imposing a known voltage in the gripped facets of the samples (see Fig. 2b). Nonetheless, this electrode does not impose an electric field perfectly aligned with the longitudinal axis of the samples. The electrical current in the inner part of the samples is hindered due to the higher resistivity in the direction perpendicular to the gripped facets. Therefore, the measured resistivity depends on the material resistivities in each principal direction (longitudinal, transverse and vertical). This measured resistivity was denoted as effective resistivity ($\rho_{eff}$). The electro-thermal characterisation was performed by imposing different DC fields (150, 187.5 and 250 V m⁻¹) while measuring the electrical resistance and surface temperature for 10 min. A DC field causes a temperature rise by Joule heating. Due to the positive dependence of the material resistivity on temperature, an increase in temperature reduces the electrical current that flows through the sample. This consumed power reduction leads to lower heat generation, proving a bidirectional coupling between the electrical and thermal responses. The results for an applied electric field of 250 V m⁻¹ are presented in Fig. 1b.1, whereas the results for the remaining DC fields are presented in Supplementary Information (Fig. S5). A temperature and effective resistivity stabilisation is found independently of the printing orientation used, occurring when the Joule heating equals the convective cooling. The Joule effect is directly related to the amount of electrical current, leading to a lower initial resistivity that provokes higher temperatures for the same applied electric field. Longitudinal samples show a preferred direction of conductive path formation coinciding with the orientation of the electric field, showing the lowest effective resistivity among the three printing configurations. Moreover, transverse samples present a preferred conductive path formation direction perpendicular to the electric field, hindering the current flow and exhibiting the highest effective resistivity. The oblique samples present a behaviour in between their counterparts. Besides their differences in effective resistivity, every printing orientation exhibits a similar stabilisation, where the printing orientation has a negligible effect on the thermal behaviour of conductive PLA components.

The strain sensitivity (referred to as the slope of the variation of the effective resistivity with the stretch) of each printing configuration was analysed via mechano-electrical tests. The resistance of each sample was measured while performing a monotonic uniaxial tensile test. These results are presented in Fig. 1b.2. Not only the initial effective resistivity depends on the printing direction (as in the electro-thermal tests), but also the variation of resistivity with deformation. Longitudinal samples exhibit a linear effective resistivity variation at stretches lower than 1.012, reaching a plateau at higher stretches. In addition, the strain sensitivity of longitudinal samples is the lowest of the three printing orientations. This can be explained by the homogeneous strain distribution in this printing configuration, applied along the filament direction. In transverse samples, the strain sensitivity is not linear and is much higher than in longitudinal or oblique configurations. The mechanical load in this configuration is applied perpendicular to the filament, causing non-homogeneous deformations, especially in zones close to the inter-filament voids. This non-homogeneous deformation explains the higher strain sensitivity. Finally, oblique samples exhibit a nearly linear response up to stretches of 1.015, with a higher strain sensitivity than their longitudinal counterparts. In this case, the loading direction is oriented at 45° with respect to the filament direction, so the response is found to be in between transverse and longitudinal configurations.

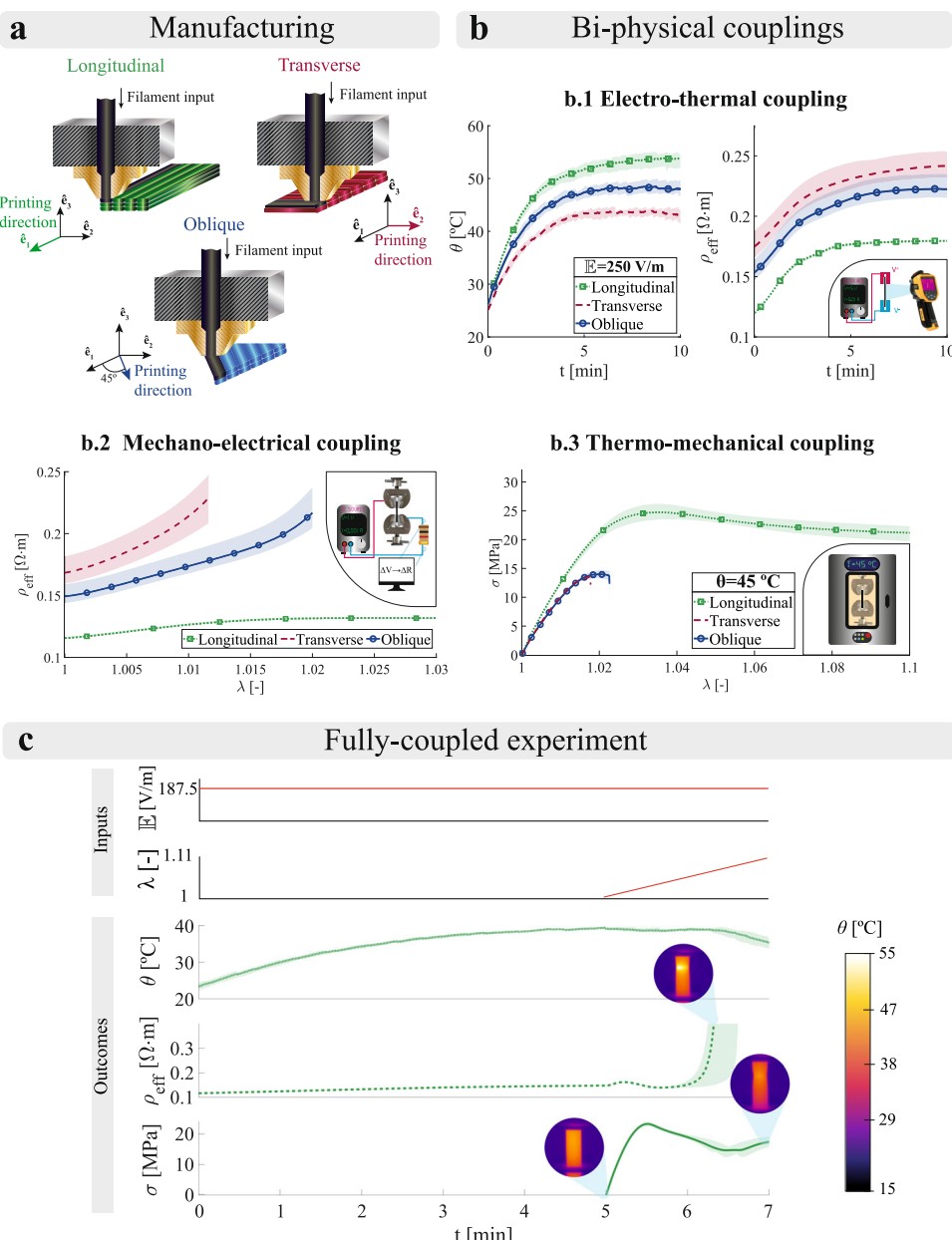

**Fig. 1 | Multi-physical characterisation of FFF samples with different printing orientations. a** Diagram of the printing directions used. **b** Experimental results isolating physics by pairs: **b.1** Electro-thermal characterisation analysing the Joule effect under an electric field ($\mathbb{E}$) of 250 V m$^{-1}$. The temperature evolution is presented on the left and the effective resistivity ($\rho_{eff}$) is presented on the right. **b.2** Mechano-electrical characterisation analysing the effective resistivity ($\rho_{eff}$) variation with tensile deformation. **b.3** Thermo-mechanical characterisation analysing the effect of the temperature ($\theta$) in the mechanical response under uniaxial tensile loading. **c** Experimental results of a fully-coupled test performed on longitudinal samples. An electric field ($\mathbb{E}$) of 187.5 V m$^{-1}$ is imposed leading to a Joule heating for 5 min and, then, a tensile stretch ramp is imposed. The average surface temperature ($\theta$), the effective resistivity ($\rho_{eff}$) and the mechanical stress ($\sigma$) are measured over time. Three replicates were used per testing condition. The lines represent the average response whereas the shaded areas represent the experimental deviation To provide fair comparisons between experimental and modelling data, as the voltage at reference configuration is kept constant during the tests, the effective resistivity and electric field are calculated in the reference configuration too.

Lastly, the use of a thermoplastic matrix introduces a strong dependence of the mechanical response on temperature. Thermomechanical tests were performed to identify the variation in mechanical properties due to temperature changes. For this purpose, uniaxial tensile tests were carried out under controlled thermal conditions, considering a range from 25 to 45 °C. The results for a testing temperature of 45 °C are presented in Fig. 1b.3. A great difference in ductility is found between longitudinal samples and transverse or oblique ones, explained by the relative loading direction with respect to the filament deposition. The inter-filament voids, for both oblique

and transverse samples, promote mechanical failure due to the debonding of adjacent filaments. Longitudinal samples present large deformations at failure that increase with higher temperatures. This is explained by the need to break the longitudinal filaments and not only the interphase between them.

To extend these pairwise tests, we performed a fully-coupled set of experiments. Using this methodology, every studied variable (i.e., temperature, electrical resistivity and displacement) was controlled and measured. The samples were placed in the universal testing machine where null displacement and an electric field of 187.5 V m$^{-1}$

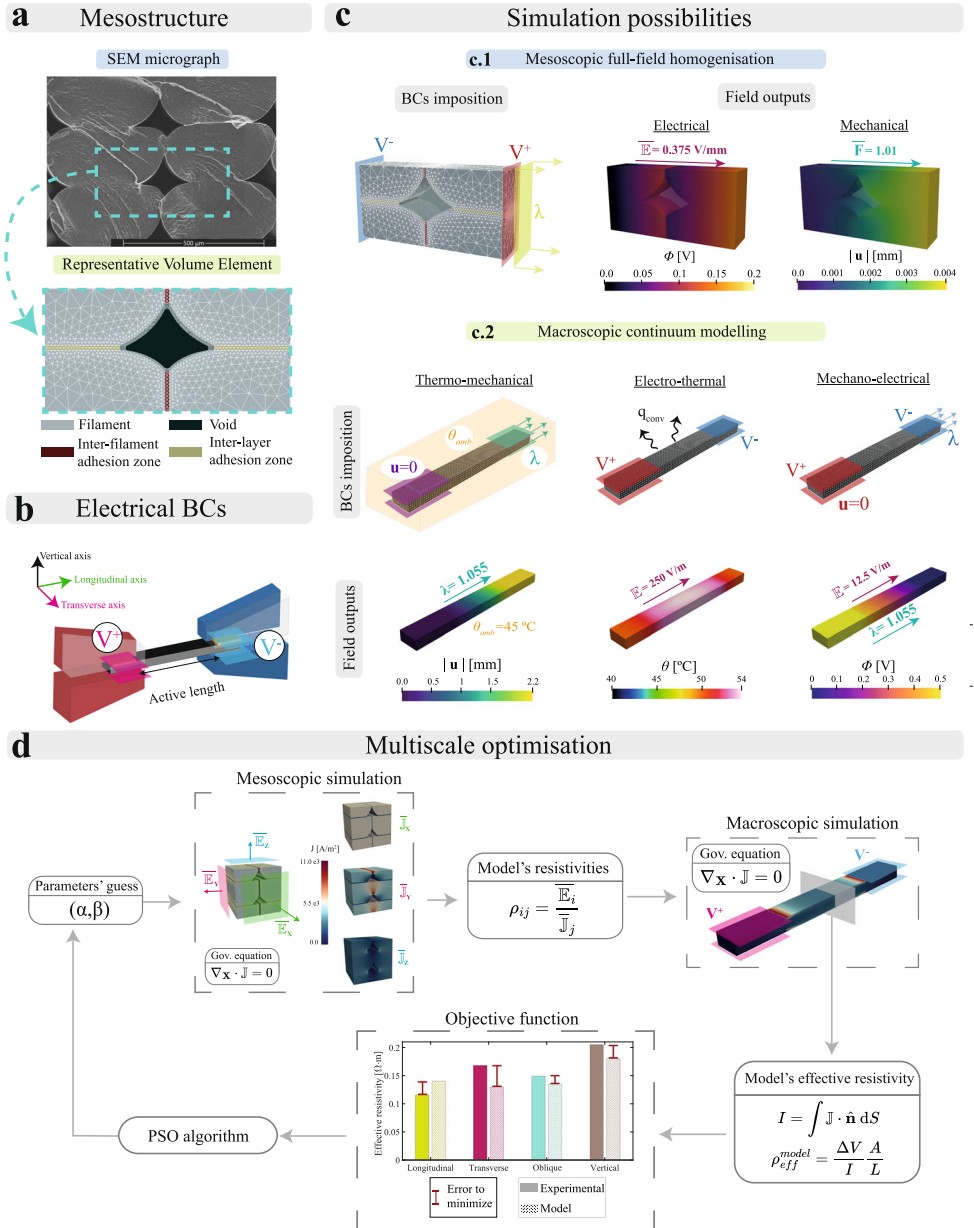

**Fig. 2 | *In-silico* multiscale platform for conductive thermoplastics. a** Scanning electron microscopy (SEM) images of longitudinal samples were used to reliably capture the mesostructural features in the representative volume elements (RVEs) used. **b** Diagram of the electrical boundary conditions (BCs) used in the printed samples. The electric potentials were applied to parallel facets of the sample using compatible gripping electrodes with the universal testing machine. **c** Examples of multiphysical simulations supported by the framework: **c.1** Mesoscopic full-field homogenisation controlling both mechanical and electrical macroscopic BCs. **c.2** Macroscopic continuum modelling controlling thermal (i.e., temperature and convective terms), electrical (i.e., electric potential) and mechanical (i.e., displacements) BCs, in independent or simultaneous manners. **d** Multiscale optimisation approach based on a Particle Swarm Optimisation (PSO) algorithm used to calibrate mesostructural and mechanical parameters. Example of application to obtain the resistivity of the interfilament adhesion zones. The homogenisation framework allows for obtaining the macroscopic resistivity in each principal direction. The macroscopic parameters feed the continuum model allowing for a minimisation of the error between numerical and experimental results (i.e., objective function).

were imposed for 5 min. After this time, a stretching ramp was applied maintaining the previous electric field. Throughout the experiment, surface temperature, effective resistivity and mechanical stress were obtained. The results for longitudinal samples are presented in Fig. 1c. The first stage of the experiment, consisting of an electro-thermal analysis, facilitated thermal and electrical stabilisation before the application of the stretching ramp. The boundary conditions during this stage impede the sample expansion in the axial direction due to electro-thermal heating. This introduces thermal stresses within the sample but these are negligible, being lower than 1% of the peak stress. Once the deformation begins, the effective resistivity initially increases and then reduces, coinciding with the beginning of the plastic regime. This reduction in effective resistivity is caused by the non-longitudinal deformations from Poisson's effect. These non-longitudinal deformations imply the approaching of conductive particles at the microscale in those directions, countering the separation in the longitudinal direction. The increase in temperature caused by Joule effect allows for an increase of the material ductility, forming significant necking. The highly localised deformation in the necking region leads to a separation of the conductive particles, high enough to break the conductive pathways in that zone and cause a loss in conductivity. Finally, the Joule heating ceases after the conductivity loss, triggering a sample cooling

that stiffens the composite, an effect observed in the increase of mechanical stress at the end of the experiment. These results evidenced the great interdependencies between thermal, electrical and mechanical contributions, as well as important effects of the printing direction in the orthotropic mechano-electrical response.

### *In-silico* platform to connect multifunctional responses to printing parameters across scales

During the printing process, mesostructural voids are formed due to imperfect adhesion between filaments. Within the adhesion layers, the polymeric chains do not entangle completely, behaving differently than the pure filament. The macroscopic performance of conductive thermoplastics is determined by such mesostructural features that present strong links with the printing process. We developed herein a multi-scale modelling framework coupling both meso- and macroscopic characteristics. That being so, it is possible to overcome the experimental difficulties needed to uncover such links. This also provides a clear roadmap on how printing parameters can be tuned to design customised multifunctional responses.

This framework combines multiscale modelling and optimisation modules. The modelling module splits the problem into two parts: (i) a full-field homogenisation framework formulated at the mesoscale; and (ii) a continuum model formulated at the macroscale. The homogenisation framework makes use of synthetically generated Representative Volume Elements (RVEs) accounting for all relevant mesostructural features, such as filament phase, inter-filament adhesion phase, inter-layer adhesion phase and mesostructural void (Fig. 2a). The homogenised formulation describes the mechano-electrical problem considering different responses for each phase (Fig. 2c.1). This framework allows for capturing the mechanical and electrical responses depending on the printing parameters and provides homogenised material parameters to feed the macroscopic continuum formulation. The macroscopic formulation accounts for: (i) a temperature dependent orthotropic elasto-viscoplastic response; (ii) a deformation dependent orthotropic electrical conductivity; (iii) Joule heating; and (iv) a transient thermal response considering convective terms. The macroscopic continuum model provides great flexibility in terms of BCs application, allowing for simulating all possible experimental conditions (Fig. 2b and c.2). Note that this methodology is not based on bottom-up homogenisation approaches[50–52], but the mesoscopic homogenisation is used to get insights that are unfeasible to capture with current experimental methods. This strategy allows to overcome the experimental difficulties needed to identify potential degradation in the multifunctional properties within the filament-to-filament adhesion regions and quantify them by direct comparison with macroscopic experimental data. More details on the formulation and its implementation can be found in Methods and Supplementary Information.

The complete modelling platform can be used to identify mesoscopic material parameters from experimental results gathered at the macroscale. The interrelation between macroscopic measurements with the mesoscopic material response is addressed by optimisation techniques, linking both scales. We proposed a multiscale optimisation approach to calibrate meso- and macroscopic parameters simultaneously, comparing the numerical solution with the macroscopic experimental results. This methodology is based on the Particle Swarm Optimisation (PSO) algorithm, finding an optimal solution within a solution space following a meta-heuristic approach. This methodology starts obtaining the desired macroscopic parameters (e.g., principal directions resistivities in Fig. 2d) by solving the homogenised problem. At this homogenisation stage, the candidate solutions are introduced as a guess for the mesoscopic constitutive parameters (e.g., inter-filament/inter-layer adhesion zones resistivities ($\alpha$, $\beta$) in Fig. 2d). The obtained macroscopic parameters are then used as an input for the continuum model, that simulates the equivalent experimental test imposing appropriate BCs. Thus, a reliable comparison between experimental and numerical results can be performed within the objective function, accounting for the effects of complex BCs that cannot be considered in the homogenised model. This procedure is performed in an iterative fashion until finding the optimal solution (see Supplementary Information for more details). This procedure was used to identify the continuum model parameters from experimental data (see experimental and numerical results in Fig. 3).

The model correctly describes the change in elastic behaviour when varying the printing orientation considering mechanical orthotropy, as well as the thermal dependence of the material's stiffness and plastic softening (Fig. 3a). Regarding the mechano-electrical predictive capabilities, the changes in both sensitivity and initial effective resistivity depending on the printing orientation were successfully captured by considering a strain-dependent orthotropic conductivity tensor (Fig. 3b). For the electro-thermal simulations, the orthotropic conductivity tensor allows to capture changes in heating depending on the printing orientation. In addition, by considering convective conditions, it is possible to capture the thermal and electrical stabilisation (Fig. 3c). Finally, the formulation was proved reliable when working with the three physics of the problem simultaneously (Fig. 3d). The model was capable of capturing: (i) the initial increase of temperature and effective resistivity induced by Joule heating; (ii) the increase of effective resistivity due to mechanical deformation and the change in the sensitivity trend when reaching plastification; (iii) the necking that leads to a loss of conductive capabilities and (iv) the cooling caused by the cease of Joule heating, with the subsequent hardening of the composite.

### Hybrid platform to optimally design multi-functional responses at the material level

Although mesoscopic structural features, such as voids, greatly impact the functional response of the 3D printed samples, these are not the only players determining their coupled thermo-electro-mechanical behaviour. The printing process also introduces changes at the material level, specifically within the inter-filament adhesion zones and the inter-layer adhesion zones. In this regard, the filament-filament fusion process generates spatial heterogeneities in the particles' distribution and modifies the material properties within the adhesion zones with respect to the internal filament phase. To account for these features, we define the geometry of each RVE by the cross section of the void and the length of the adhesion zones, thus extending the previous *in-silico* platform. To this end, the RVEs distinguish four phases: i) printed filament; ii) inter-layer adhesion zone; iii) inter-filament adhesion zone; and iv) mesostructural void. The changes in material properties are modelled by applying a degradation coefficient to the adhesion phases. A set of initial degradation parameters describes the response of the adhesion zones based on the filament behaviour. This approach can predict the behaviour when modifying printing parameters that have a direct impact on the mesostructure (i.e., layer width, layer height or printing orientation). To study all the possibilities that this methodology offers, mechano-electrical simulations were performed in several RVEs changing geometrical and material features. The mechano-electrical conditions are applied by imposing both a macroscopic stretch and an electric field, emulating the boundary conditions applied in the experimental campaign.

Figure 4a presents simulations for longitudinal and transverse configurations comparing RVEs with and without adhesion zones. The consideration of adhesion zones (softer and less conductive than the filament phase) provokes an increase in resistivity, especially in non-longitudinal configurations. The simulations show that such considerations are essential to reproduce the experimental difference observed in conductivity and strain sensitivity between longitudinal and transverse samples. In addition, the non-linear behaviour in resistivity variation for transverse samples is captured considering the

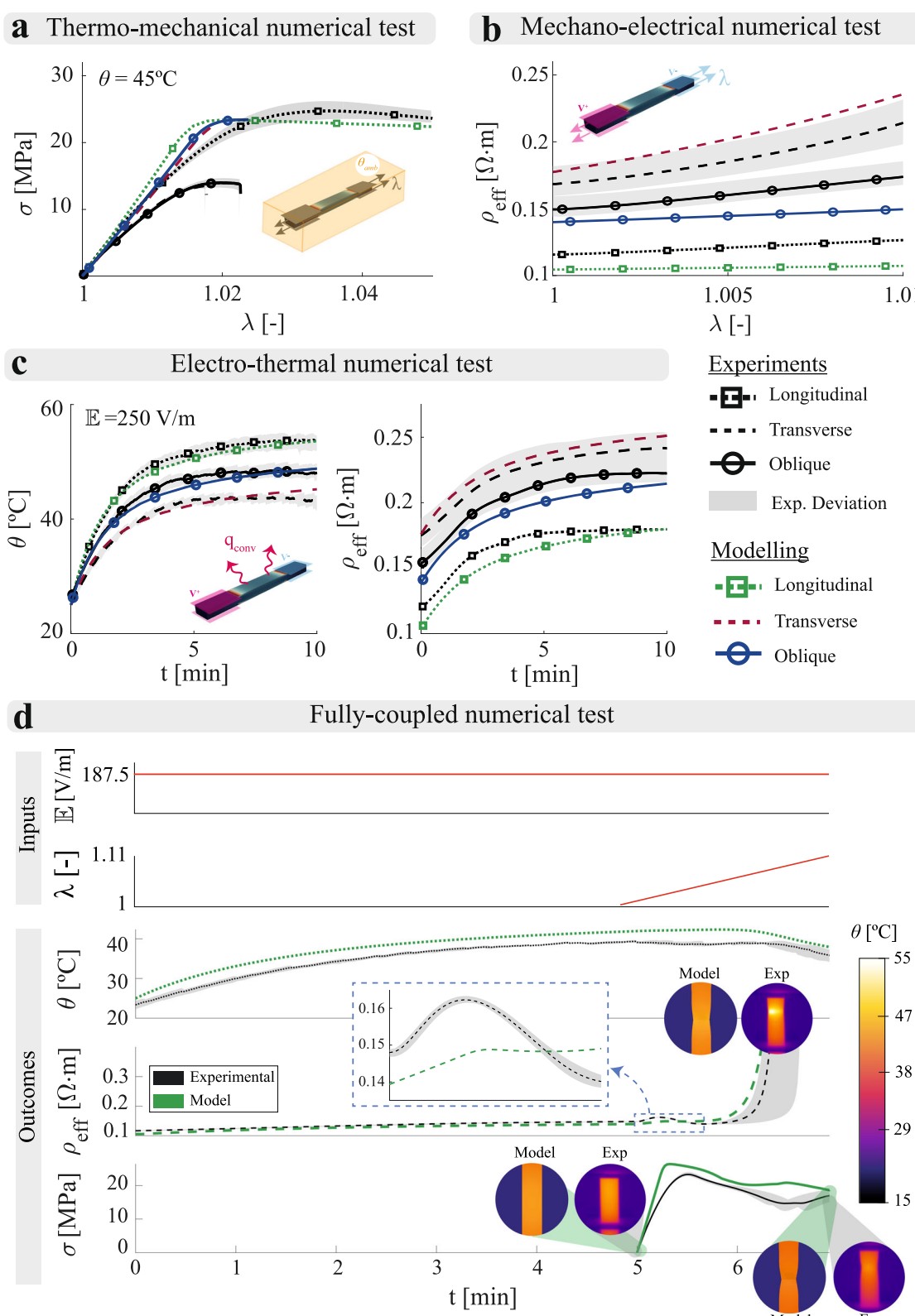

**Fig. 3 | Predictive capabilities of the multi-physical *in-silico* platform.**
**a** Simulations of thermo-mechanical tests at a temperature of 45 °C on longitudinal, transverse and oblique samples. **b** Simulations of mechano-electrical tests on longitudinal, transverse and oblique samples. **c** Simulations of electro-thermal tests applying an electric field ($\mathbb{E}$) of 250 V/m on longitudinal, transverse and oblique samples. **d** Simulation of a thermo-electro-mechanical test applying an electric field ($\mathbb{E}$) of 187.5 V/m and uniaxial tensile loading on a longitudinal sample. Experimental (black) and numerical (blue/green/red depending on the printing orientation) results are presented together in all graphs. The grey shaded areas represent the experimental deviation of the tests. To provide fair comparisons between experimental and modelling data, as the voltage at reference configuration is kept constant during the tests, the effective resistivity and electric field are calculated in the reference configuration too.

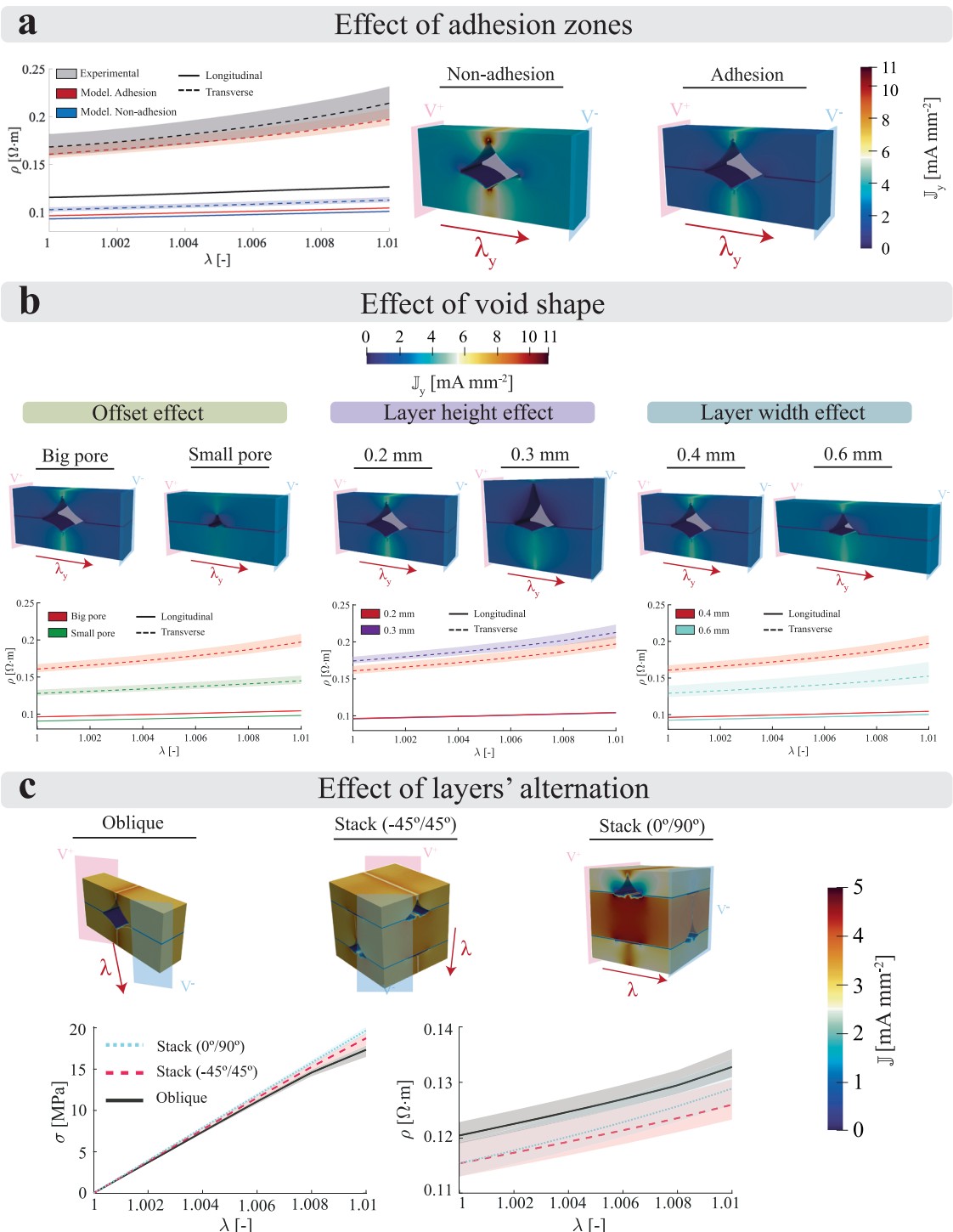

**Fig. 4 | Full-field homogenisation analysis of 3D printed conductive thermoplastics at the mesoscale. a** Effect of considering adhesion zones in the RVEs. Both experimental and numerical results of mechano-electrical tests for longitudinal and transverse samples are presented, comparing the consideration and absence of adhesion zones. Two simulated electrical current densities ($\mathbb{J}$) for transverse samples are shown at the maximum deformation stage. **b** Predictions of the mechano-electrical response of 3D printed conductive thermoplastics depending on void's geometry. The simulations evaluated the coupled responses depending on the height of the void, the layer height, and the layer width. **c** Predictions of the mechano-electrical response of 3D printed conductive thermoplastics depending on the layers' alternation, considering: i) oblique (45°); ii) stack oblique (± 45°); and iii) stacked directions (0°/90°). The experimental data refers to tests conducted on PLA/CB samples.

same mechano-electrical constitutive relation and parameters for adhesion zones and filament. Figure 4b presents the dependence of the mechano-electrical response with the form and size of voids. Three variations are considered, simulating the change of different printing parameters: (i) form of the voids' cross section, emulating the effect of

the printing offset; (ii) height of the voids, emulating the effect of layer height; and (iii) width of the voids, emulating the effect of layer width (nozzle diameter).

The effect of the printing offset, defined as the distance between the nozzle and the printing bed during the printing of the first layer,

has been studied as it is one important mechanism governing void shape. Figure 4b shows that a bigger pore leads to both more resistive and strain sensitive behaviour. The material behaves as more resistive due to the reduction of area for the current to flow. A smaller adhesion area leads to an increase in strain concentrations and, therefore, a higher change in electrical resistivity in those zones. The transverse configuration is more affected by mesostructural changes than its longitudinal counterpart. The layer height is one of the most commonly modified parameters in 3D printing technologies, reducing the printing time in exchange of printing quality. In terms of the layer height, its increase leads to larger voids, increasing mainly their height[53]. To analyse this phenomenon, two different RVE types were used to simulate a mechano-electrical problem, considering a layer height of 0.2 mm and 0.3 mm. Figure 4b shows that, in the case of transverse configuration, the inter-filament area is reduced for the 0.3 mm layer height RVE, leading to higher resistivity values. To predict the mechano-electrical behaviour of samples printed with different nozzles, two different RVE types were used fixing the void's size and form, but modifying the width of the RVE, emulating layer's widths of 0.4 mm and 0.6 mm. The aspect ratio between void and filament phases is reduced for 0.6 mm layer width. Hence, the conductivity of the longitudinal configuration is slightly enhanced with respect to the 0.4 mm width RVE. For the transverse configuration, the conductivity properties are also enhanced while maintaining a similar strain sensitivity.

A final study analyses the dependence of the mechano-electrical response with the layers' alternation. Three RVE forms are considered: (i) oblique (45°); ii) stack oblique (± 45°); and iii) stack (0°/90°), which presents overlapping of longitudinal and transverse layers. Both stress-stretch and resistivity-stretch curves are presented in Fig. 4c along with the current density results at the final time step. In the oblique configurations, concentrations of current flow density appear in the vicinity of the inter-filament adhesion zones, whereas in the interior of those zones, the current flow is considerably lower. In the case of the stack configuration (0°/90°), the maximum amount of current flow appears along the longitudinal layers, where no inter-filament or void opposition to the current flow is found, whereas a concentration of current is created in the inter-filament zones along the transverse layers. Regarding the average response of each configuration, several aspects can be highlighted. The higher stiffness is found in stack configurations since half of the RVE is facing the load in a longitudinal manner, followed by stack oblique and oblique, in which the inter-filament zone supports a high amount of stress. The initial resistivity value for each configuration is similar, especially for oblique and stack arrangements. The resistivity strain sensitivities also present similar tendencies between printing directions.

## Smart selector of printing parameters to design multi-functional structural components

The previous mesostructural analysis highlights the printing direction as the manufacturing parameter that can modulate most of the multi-functional behaviour of the material. We propose here the use of the *in-silico* platform to first identify the ideal distribution of material properties at the macroscale to deliver an optimal multi-functionality and, then to optimise the model parameters that capture this spatial material distribution at the mesoscale. This parametric outcome can finally be used to establish the printing strategy resulting in the optimal structural design of a functional component. We tested the feasibility of this idea by implementing a proof of concept case. To this end, we propose the optimal manufacturing of a heating cartridge for a direct ink writing (DIW) printer. DIW printers are based on extrusion techniques to selectively deposit a viscous polymeric ink by layers. For specific inks, it is needed to enhance their viscosity before extrusion[22,54]. In some cases, this can be achieved by including additives in the mixture or by increasing their temperature to accelerate

the polymerisation process. The designed component aims to surround the syringe that contains the printable ink and heat it to desirable temperatures by Joule heating using a conductive thermoplastic component (Fig. 5a). The heating capabilities of the cartridge are defined not only by geometrical features and electric BCs but also by the inherent conductivity orthotropy of PLA/CB components. The preferred conductive directions within the print can lead to unwanted localisation of electric potential and, therefore, to a nonuniform temperature profile within the component. This can be crucial, leading to high temperatures (e.g., above glass transition temperature) in certain zones while not reaching high enough temperatures in the rest of the component. In Fig. 5b we show the temperature profile of two cartridges with the same geometry but different printing orientations, heated with the same electric potential difference. Both experimental and numerical results (using our continuum modelling framework) are presented. The first printing strategy (i.e., printing direction along the longitudinal axis of the component) reaches non-desirable temperatures in the electrode zones and the area near the syringe nozzle, with considerably lower temperature in the middle zone. Moreover, the second printing strategy (i.e., printing direction perpendicular to the longitudinal axis of the component) reaches, on average, lower temperatures compromising the optimal heating of the ink.

The problem at hand, therefore, consists of providing the spatial distribution of printing directions that reduces the heterogeneity in Joule heating when applying a specific electric actuation. To tackle this, we posed the optimisation problem. To this end, the cartridge geometry was divided into three partitions presenting special features. These correspond to the different coloured volumes in Fig. 5c: (i) the area where the voltages are imposed, using the bolts that attached the cartridge to the printer as electrodes; (ii) the middle section; and (iii) the end of the syringe. An analogous optimisation algorithm to the one presented in Fig. 2d was employed to obtain the optimal printing direction in each section. The angles to optimise were: the inclination angle of the cartridge with respect to the printing bed $\alpha_X$, which is equal for every partition; and the printing orientation of each partition $\alpha_Z^i \quad \forall i \in 1, 2, 3$. These angles were used as parameter inputs when modelling the electro-thermal heating within the cartridge. The optimisation algorithm (PSO) searched for printing orientations leading to the lower standard deviation of the nodal temperatures. The resulting experimental and numerical temperature profiles of the optimised cartridge are shown in Fig. 5d. The optimised cartridge was then used to correctly print a magneto-responsive device. As shown in Fig. 5e, the use of the cartridge allowed the printing of the ink without scattering, overcoming the existing limitations when printing the ink at room temperature.

## DISCUSSION
This work provides a hybrid, i.e., experimental and computational, platform to optimise the multi-functional design of polymeric components manufactured by filament extrusion techniques. The main challenge associated with this manufacturing process is the highly dependent thermo-electro-mechanical performance of these materials on mesostructural features determined by the printing strategy. Since the analysis of every printing parameter affecting the thermo-electro-mechanical behaviour is experimentally unapproachable, we introduced an *in-silico* platform that connects the material mesostructure with the macroscopic behaviour by combining different modelling approaches. To build the hybrid framework and demonstrate its validity, we took a conductive thermoplastic (PLA/CB) as reference. First, we identified the interplays between electrical, thermal and mechanical responses making use of the experimental framework. This experimental platform tackled the thermo-electro-mechanical problem by isolating the physics by pairs and ultimately performing fully-coupled tests. Samples with different printing orientations were tested, as it was identified as the printing parameter with the most

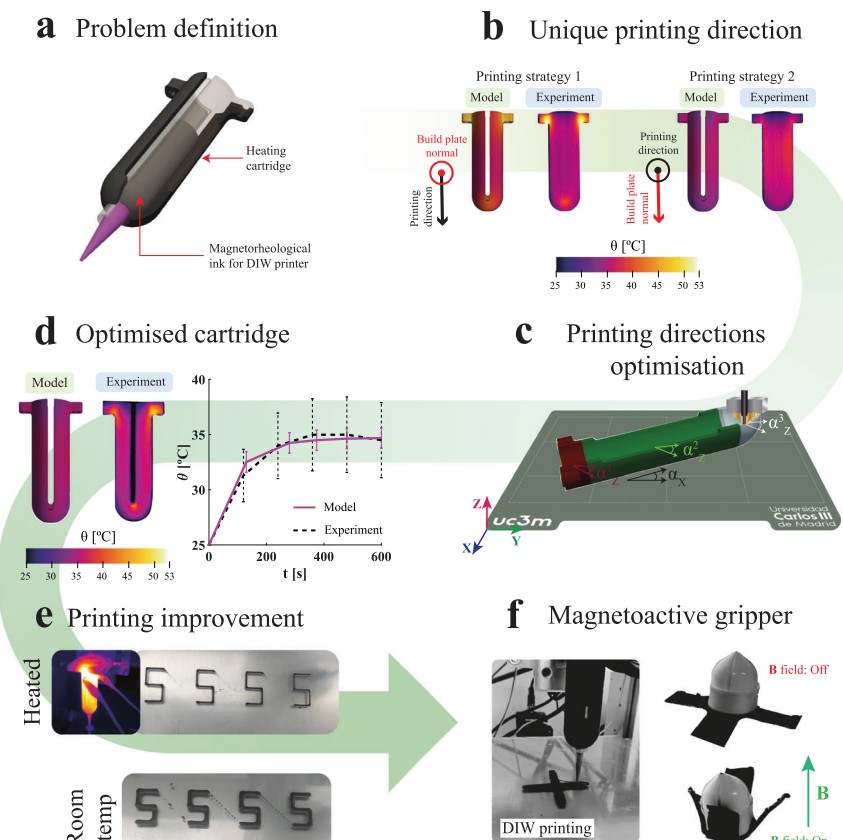

**Fig. 5 | Application of the *in-silico* platform to the optimal design of a 3D printed heatable cartridge for DIW printers. a** Design of a 3D printed heatable cartridge used to enclose a conventional syringe containing a magnetorheological ink for DIW printers. Controlled increases in temperature improve the printability of the ink enhancing its polymerisation. **b** Temperature distribution measured after the application of 30V for 10 min, showing numerical and experimental results for two printing strategies. **c** Optimised solution provided by the *in-silico* platform in terms of local printing directions to provide a uniform distribution of temperatures. The inclination angle with respect to the printing bed is defined by $\alpha_X$, while the proper printing direction of each cartridge division ($i$) is defined by $\alpha_Z^i$. **d** Comparison of the temperature profile obtained experimentally and numerically from the optimised heatable cartridge. The average surface temperature is represented in the graph along with its standard deviation. **e** Comparison of the experimental printing resolution improvement when using the heating cartridge. **f** Printing of a magnetoactive gripper. This gripper closes when a magnetic field is applied.

important effect on the multi-functional response[34]. The multiscale modelling platform was then used to predict the multifunctional responses of the components at the mesoscale, using a full-field homogenisation formulation, and at the macroscale, using continuous modelling formulation. Ultimately, we illustrated the potential of the whole framework to address a real technological challenge, i.e., the optimisation of a structural component for a DIW printer with the capability to act as an electro-heater.

The experimental results showed a strong dependence of both conductive and mechanical responses on printing orientation. These responses are clearly enhanced when the electro-mechanical loading is aligned with the filament deposition direction. Contrary, when the loading is applied transversely to the filament direction, there is an increase in brittleness and electric resistivity. These observations suggest the existence of an orthotropic mechano-electrical nature with filament deposition, inter-filament adhesion and inter-layer adhesion directions as the principal directions. This orthotropy does not only affect isolated physical behaviours, but it also governs the coupled responses determining the mechano-electrical interplay. This dictates drastic changes in trends due to different strain sensitivities in each principal direction.

The multiscale computational platform developed predicts the multi-functional behaviour of the conductive components at the macroscale from their material and mesostructural characteristics. This point is very relevant since the material features are determined

by the filament used, and the mesostructural features can be modulated by the printing strategy chosen. First, we used the mesostructural approach to numerically demonstrate how, from a material perspective, an imperfect adhesion process during printing hinders the conductivity capabilities in those zones. By knowing the conductivity loss in the adhesion zones, the uncertain response when modifying printing parameters relies only on structural features. Accounting for this, we were able to predict the response of printed samples when varying printing parameters such as layer height, layer width or printing direction. Making use of homogenisation techniques, the macroscopic modelling tool can be fed with parameters derived from the mesoscopic analysis while considering complex BCs at the macroscale. The continuum modelling framework demonstrated its capability to correctly predict the orthotropic multi-physical behaviour of conductive printed PLA/CB, capturing the main phenomena observed in the experimental campaign. These frameworks, in combination with optimisation techniques, were used to optimise the printing strategy for a heating cartridge used in a DIW printer. The optimisation problem was focused on the spatial selection of the printing direction to provide a distribution of material conductivity so that the structural component is heated as homogeneously as possible under electric actuation. The computational solution was experimentally validated showing an excellent performance of the approach and opening exciting opportunities for future multi-functional design of structural components.

## Methods

### Material and manufacture

Conductive PLA/CB filament from ProtoPasta (WA, USA) was employed to fabricate both the rectangular samples and the heating sockets. Every print was manufactured using a Prusa i3 MK3S+ (Prusa Research, Czech Republic) with a hardened-steel nozzle with a diameter of 0.4 mm. The rectangular samples had a cross section of 10 × 5 mm and a length of 80 mm. The main set of printing parameters was: layer width of 0.4 mm, layer height of 0.2 mm, extruding temperature of 230 °C, printing bed temperature of 60 °C, velocity of 30 mm/s and 100 % of infill. The magnetorheological ink used to validate the heating socket application was fabricated by mixing PDMS (DOW Sylgard 184) with hard magnetic particles (MQP-S-11-9-grade powder, Neo Materials Technology Inc., Greenwood Village, Colorado, United States). The concentration of particles within the composite was 30 vol %. The ink was then transferred to a cartridge compatible with an *in-house* DIW printer[22].

### Experimental procedure for electro-thermal tests

As also described in[26], an adjustable DC power source was used in the experiments to supply a constant DC voltage ($V_o$) to the samples. Ad-hoc electrode grips were manufactured and attached to the universal testing machine to ensure correct electrical contact and a consistent distance between electrodes (L). The distance between electrodes was set to 40 mm. Three voltages were considered: 5 V, 7.5 V and 10 V leading to the corresponding electrical fields of 125 V/m, 187.5 V/m and 250 V/m, respectively. The temperature of the samples was recorded by an infra-red camera (FLUKE TiS75+, WA, USA). The effective resistivity of the samples ($\rho_{eff}$) was obtained by measuring the resistance of the samples ($R_i$) using a data-acquisition system (DATAQ DI-2008, OH, USA). As the data-acquisition system only measures voltage values, the material resistivity was calculated via voltage drop ($V_{drop}$) on an additional 1 Ω resistor ($R_{ref}$) connected in series. This resistor presented a negligible value compared to the samples' resistance to not interfere with the current causing Joule heating. The relation used to calculate the samples' effective resistivity from the measured voltage was the following:

$$\rho_{eff} = \frac{R_i A}{L} \quad ; \quad R_i = \frac{R_{ref}(V_o - V_{drop})}{V_{drop}} \tag{1}$$

where A is the area of the sample's cross-section and $V_o$ the applied voltage.

### Experimental procedure for thermo-mechanical tests

Following the procedure described in[26], uniaxial tensile tests were performed on rectangular samples at a strain rate of 0.001 s⁻¹ (INSTRON 34TM-5, MA, USA) and using a climatic chamber (INSTRON 3119-605, MA, USA) to control the experiment temperature. A load cell of 5 kN was used, along with wedge action grips of 5 kN. These tests were performed at different testing temperatures of 25, 35, 45 °C.

### Experimental procedure for mechano-electrical tests

Uniaxial tensile tests were performed using the ad-hoc electrode grips employed for the electro-thermal testing while measuring the samples' resistance. The resistance values were measured using the same approach detailed in Methods for electro-thermal tests. In this case, the reference in-series resistor had a value 465 Ω, to improve the voltage divisor sensitivity. The tests were performed at a controlled strain rate of 0.001 s⁻¹. A pre-compression at the gripping regions was applied to ensure proper electrical contact between the sample and the electrodes. We experimentally defined this gripping condition by the minimum pre-compression enabling stable electric conduction and avoiding sample sliding during the tensile test. We further evaluated potential excessive stress concentration at the gripping regions by confirming that the samples break occurs far from these regions.

### Experimental procedure for thermo-mechano-electrical tests

A combination of the experimental setups presented above, namely the electro-thermal and mechano-electrical tests, was used. Following a similar protocol to previous works[26], these experiments consisted of two phases: (i) electro-thermal test in which a constant electric field of 187.5 V/m was applied to filament samples for five minutes leading to a temperature and resistivity stabilisation; and (ii) uniaxial tensile test at a controlled strain rate of 0.001 s⁻¹ while maintaining the electric field. The resistivity was measured using an in-series resistor of 1 Ω to obtain more accurate measures of the current that causes Joule heating.

### Constitutive and computational framework of the full-field homogenised model

The mesostructural computational framework aims to determine the homogenised material behaviour. To this end we followed a similar methodology to[26], solving the electro-mechanical equilibrium at the mesoscopic level by applying macroscopic loads to a mesostructural domain[55,56]. This is achieved by generating a mesoscopic domain through the periodic repetition of a unit cell (RVE), which represents the smallest sample of heterogeneity within the mesostructure. For FFF printed samples with 100 % infill, the RVE includes the mesostructural void formed due to the imperfect fusion between adjacent filaments, as well as a portion of the four adjoining filaments. This geometrical representation is enforced to be periodic and, in combination with periodic boundary conditions, assumes an idealised perfect infinite arrangement of deposited filaments. Then, the full-field coupled electro-mechanical problem is posed and solved in a periodic fixed domain defined in the reference configuration where four phases are considered: the filaments, the void, the inter-layer adhesion zone and the inter-filament adhesion zone. Therefore, the problem field variables to be solved are the displacement field, **u** and the electric potential field, $\phi$.

Within the RVE, the mechanical response of the different phases is assumed to behave as an elastic-viscoplastic solid. The proposed rheological model consists of a unique branch, formed by a hyper-elastic spring in series with a viscoplastic dashpot. The Piola-Kirchhoff stress is defined by a compressible Neo-Hookean model as

$$\mathbf{P}(\mathbf{F}, \mathbf{F}^p) = [\mu(\mathbf{B}^e - \mathbf{I}) + \lambda ln(J_F^e)\mathbf{I}] \cdot \mathbf{F}^{-T} \tag{2}$$

where $\mu$ and $\lambda$ are the Lamé constants, $\mathbf{B}^e = \mathbf{F}^e(\mathbf{F}^e)^T$ and $J_F^e = det(\mathbf{F}^e)$. The plastic deformation follows the multiplicative decomposition of the deformation gradient into elastic, $\mathbf{F}^e$, and plastic, $\mathbf{F}^p$, components; $\mathbf{F} = \mathbf{F}^e \mathbf{F}^p$. The plastic behaviour is dictated by the isotropic yield surface

$$f = \overline{\sigma}_{VM} - \sigma_Y - k \tag{3}$$

where $\overline{\sigma}_{VM}$ is the Von Mises equivalent stress and $\sigma_Y$ and k are the yield stress and the isotropic hardening (Voce law), respectively. The associative plastic flow reads

$$\dot{\overline{\varepsilon}}^p = \begin{cases} \dot{\varepsilon}_o \left( \exp[\frac{1}{C}(\frac{\overline{\sigma}_A}{\sigma_Y + k}) - 1] - 1 \right) & f > 0 \\ 0 & f <= 0 \end{cases} \tag{4}$$

where $\dot{\varepsilon}_o$ is the reference strain rate and $C$ is the strain-rate sensitivity parameter[57]. The plastic flow direction follows the deviatoric part of the Cauchy stress $\mathbf{N} = \sqrt{3/2}\, \boldsymbol{\sigma}_A^{dev} / \| \boldsymbol{\sigma}_A^{dev} \|$, defining the rate of plastic deformation gradient as $\dot{\mathbf{F}}^p = (\mathbf{F}^e)^{-1} \cdot \dot{\overline{\varepsilon}}^p \mathbf{N} \cdot \mathbf{F}$. The field equation for the mechanical equilibrium is set as a linear momentum balance under the

absence of body forces, and defined in the reference configuration as

$$\nabla_{\mathbf{X}} \cdot \mathbf{P} = 0 \tag{5}$$

where $\nabla_{\mathbf{X}}$ is the gradient operator with respect to the reference configuration.

Regarding the electric problem within the RVE, all phases provide a linear relation between electric current density ($\mathbb{j}$) and electric field, which can be expressed in the current configuration as

$$\mathbb{j} = \Sigma \mathbb{e} \tag{6}$$

where $\mathbb{e}(\mathbf{x}) = -(\mathbf{F}^{-1})^{T} \cdot \nabla_{\mathbf{X}} \phi(\mathbf{X})$ is the electric field in the current configuration ($\phi(\mathbf{X})$ is the electric potential), and $\Sigma$ is the electrical conductivity. Thus, the electric current density as a function of the electric field $\mathbb{E}(\mathbf{X})$ in the reference configuration reads

$$\mathbb{J} = J_F \mathbf{F}^{-1} \mathbb{j} = \Sigma J_F \mathbf{F}^{-1} \cdot (\mathbf{F}^{-1})^{T} \cdot \mathbb{E} . \tag{7}$$

The governing equations for the electrical stationary problem correspond to the Maxwell equations in the absence of magnetic fields:

$$\nabla_{\mathbf{X}} \cdot \mathbb{D} = Q \quad ; \quad \nabla_{\mathbf{X}} \times \mathbb{E} = 0 \quad ; \quad \nabla_{\mathbf{X}} \cdot \mathbb{J} = 0 \tag{8}$$

with $Q$ being the charge accumulated within the body and $\mathbb{D}$ the electric displacement. Note that the third equation is obtained by applying the divergence operator to Ampere's circuital law and assuming $\nabla_{\mathbf{X}} \cdot \epsilon_0 \frac{\partial \mathbb{E}}{\partial t} = \frac{\partial Q}{\partial t} = 0$. To account for the modification of conductive paths within the filament and inter-layer/inter-filament adhesion zones, $\Sigma$ has been proposed linearly dependent with the first invariant of the deformation gradient ($I_1^F$):

$$\Sigma(\mathbf{F}) = \Sigma_{ref}\left(1 - C_F(I_1^F - 3)\right), \tag{9}$$

with $\Sigma_{ref}$ as the conductivity at an undeformed state and $C_F$ as a strain sensitivity parameter.

To obtain the macroscopic effective behaviour and by applying homogenisation concepts, the total displacement field, $\mathbf{u}(\mathbf{X})$, can be decomposed into a macroscopic variation and a fluctuating displacement field $\widetilde{\mathbf{u}}(\mathbf{X})$ as

$$\mathbf{u}(\mathbf{X}) = (\overline{\mathbf{F}} - \mathbf{I}) \cdot \mathbf{X} + \widetilde{\mathbf{u}}(\mathbf{X}). \tag{10}$$

Similarly, the electric potential field $\phi$ can be decomposed into a macroscopic contribution and a fluctuating electric potential field $\widetilde{\phi}$ as

$$\phi = -\overline{\mathbb{E}} \cdot \mathbf{X} + \widetilde{\phi} . \tag{11}$$

The fluctuating variables are implicitly solved by the governing field equations, resulting in the macroscopic behaviour as a function of the macroscopic test inputs.

## Constitutive and computational framework of the continuum model

The continuum macroscopic framework aims to capture the orthotropic thermo-electro-mechanical response of the 3D printed PLA/CB composite at the macroscopic scale. Due to the diverse printing orientations considered, this framework will make use of rotation tensors to work in a reference orientation configuration ($\hat{\Omega}$), i.e., longitudinal orientation. As for the homogenised approach, this formulation is set considering finite deformations.

Mechanically, the parts are considered to behave as elasto-viscoplastic solids, with an orthotropic hyperelastic response. Thus, the Piola-Kirchhoff stress is defined by an orthotropic compressible

Saint Venant-Kirchhoff model[58,59] as

$$\mathbf{P} = \mathbf{F} \cdot \mathbf{S} = \mathbf{F}^{e} \cdot \left( \sum_{i,j}^{3} \lambda_{ij}(\theta)\, \mathrm{tr}(\mathbf{E}^{e} \cdot \mathbf{L}_{jj})\mathbf{L}_{ii} + 2\sum_{i,j\neq i}^{3} \mu_{ij}(\theta)\mathbf{L}_{ii} \cdot \mathbf{E}^{e} \cdot \mathbf{L}_{jj} \right) \cdot \left((\mathbf{F}^{p})^{-1}\right)^{T}, \tag{12}$$

where $\mathbf{E}^{e} = \frac{1}{2}(\mathbf{C}^{e} - \mathbf{I})$ is the elastic Green-Lagrange strain tensor, $\lambda_{ij}$ are the Lame's constants in each direction, $\mu_{ij}$ are the shear modulus in each direction, and $\mathbf{L}_{ii}$ are the so called structural tensors that allow to define principal directions within the model. The modelling of the plastic behaviour has been defined in the homogenisation section, see Eqs. (3)–(4).

In terms of the electrical response of the printed composite, a clear orthotropy was observed during the experimental campaign. Thus, the conductivity has been defined as a diagonal tensor gathering the conductivity responses at each principal direction (i.e., $\mathbf{l}_1$, the direction along filament; $\mathbf{l}_2$, the direction between adjacent filaments; $\mathbf{l}_3$, building direction):

$$\hat{\boldsymbol{\Sigma}} = \begin{bmatrix} \hat{\Sigma}_{11} & 0 & 0 \\ 0 & \hat{\Sigma}_{22} & 0 \\ 0 & 0 & \hat{\Sigma}_{33} \end{bmatrix}. \tag{13}$$

As observed in compressive mechano-electrical experimental results (see Supplementary Information), there is a maximum conductivity that this material can reach at each direction ($\Sigma_{ii}^{crit}$). In addition, when the particles at the microscale are sufficiently separated to stop tunnelling effect, there is a cease of conductive properties, i.e., $\Sigma = 0\,S/m$. As the conductive properties vary with strain between these two values, we proposed sigmoid-based functions to describe the conductivity variation at each principal direction. Following this experimental evidence, the conductivity along $\mathbf{l}_1$ (i.e., filament direction) is proposed dependent with the square root of the fourth invariant of the right Cauchy-Green tensor along longitudinal direction ($\sqrt{I_4^1}$) as well as with the constriction deformation ($dA/dA_o$)[60]. The former variable determines the deformation along the longitudinal direction, projecting the right Cauchy-Green tensor ($\mathbf{C} = \mathbf{F}^{T} \cdot \mathbf{F}$) with the longitudinal structural tensor ($\mathbf{L}_{11} = \mathbf{l}_1 \otimes \mathbf{l}_1$); $I_4^1 = \mathbf{C} : \mathbf{L}_{11}$. The latter variable determines the deformation caused by Poisson's effect in the directions perpendicular to the longitudinal one; $dA/dA_o = J_F \sqrt{(\mathbf{F}^{-1})^{T} \cdot \mathbf{l}_1 \cdot (\mathbf{F}^{-1})^{T} \cdot \mathbf{l}_1}$. This conductivity is set as

$$\hat{\Sigma}_{11} = \frac{a_{11}(\Sigma_{11}^{crit}, \Sigma_{11}^{\theta})}{b_{11}(\Sigma_{11}^{crit}, \Sigma_{11}^{\theta}) + exp\left(C_{11}^{F}(\sqrt{I_4^1} - 1) - C_{dA}^{F}(1 - dA/dA_o)\right)}, \tag{14}$$

where $C_{11}^{F}$ is the sensitivity parameter related to the longitudinal deformation and $C_{dA}^{F}$ is the sensitivity parameter related to the constriction deformation.

In the case of the remaining principal directions $\mathbf{l}_2$ (i.e., adjacent filaments or transverse) and $\mathbf{l}_3$ (i.e., building direction or vertical), their conductivity is set to be dependent with the fourth invariant of the right Cauchy-Green tensor along the studied direction as:

$$\hat{\Sigma}_{ii} = \frac{a_{ii}(\Sigma_{ii}^{crit}, \Sigma_{ii}^{\theta})}{b_{ii}(\Sigma_{ii}^{crit}, \Sigma_{ii}^{\theta}) + exp(C_{ii}^{F}(\sqrt{I_4^i} - 1))} \quad \forall i \in \{2, 3\}, \tag{15}$$

where $C_{ii}^{F}$ is the sensitivity parameter related to the $i^{th}$ deformation. As in Eq. (14), the variables $a_{ii} = \Sigma_{ii}^{\theta}\Sigma_{ii}^{crit}/(\Sigma_{ii}^{crit} - \Sigma_{ii}^{\theta})$ and $b_{ii} = \Sigma_{ii}^{\theta}/(\Sigma_{ii}^{crit} - \Sigma_{ii}^{\theta})$ define the conductivity without any deformation ($\Sigma_{ii}^{\theta}$) as well as the maximum conductivity allowed for each direction ($\Sigma_{ii}^{crit}$).

Note that along the formulation of the different principal directions conductivities, the variable $\Sigma_{ii}^{\theta}$ has been used. This variable captures the conductivity dependence with temperature, and it is set as:

$$\Sigma_{ii}^{\theta} = \Sigma_{ii}^{ref}\left(1 - \alpha_T(\theta - \theta^{ref})\right), \tag{16}$$

where $\Sigma_{ii}^{ref}$ is the conductivity at $i^{th}$ direction at the reference temperature (25 °C). In addition to the conductivity dependence with temperature, the mechanical response is also altered by the thermal state. In this regard, the Young's modulus, yield stress and hardening parameter are proposed to decrease with temperature as follows:

$$\sigma_Y(\theta) = \sigma_Y^{ref} - C_Y(\theta - \theta^{ref})^2 \; ; \; E_i(\theta) = E_i^{ref} - C_E(\theta - \theta^{ref})^2 \; ; \; H(\theta) = H^{ref} + C_H(\theta - \theta^{ref}) \tag{17}$$

The governing equation for the thermal problem is set as a transient heat balance accounting for convective terms and a dissipative heat generation from Joule effect as

$$\varrho\dot{\theta} - \nabla_{\mathbf{X}} \cdot (\kappa J_F \mathbf{F}^{-1} \cdot \mathbf{F}^{-T} \cdot \nabla_{\mathbf{X}}\theta) - \mathbb{J} \cdot \mathbb{E} + h(\theta - \theta^{ref}) = 0, \tag{18}$$

with $\varrho$ as the specific heat capacity multiplied by the density, $\kappa$ as the thermal conductivity and $h$ as the convective parameter.

## Data availability

All data generated and analysed during this study is provided in the Source Data file as well as in the Supplementary Information. The raw data are available from the corresponding author upon request. Source data are provided with this paper.

## Code availability

The FEniCS codes developed in this study are available in the Code Ocean database (https://doi.org/10.24433/CO.1003020.v1).

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

## Acknowledgements

The authors acknowledge Prof. Marc-Andre Keip for his advice in relation with the numerical formulation. JCM, SGH and DGG acknowledge support from the 2024 Leonardo Grant LEO24-1-12283-ING-ING-13 for Scientific Research and Cultural Creation from the BBVA Foundation. The BBVA Foundation accepts no responsibility for the opinions, statements and contents included in the project and/or the results thereof, which are entirely the responsibility of the authors. The authors acknowledge support from MCIN/AEI/10.13039/501100011033 under Grant number TED2021-129709B-I00, and from the European Union NextGenerationEU/PRTR. SGH acknowledges support from the Talent Attraction grant (CM 2022 - 2022-T1/IND-23971) from the Comunidad de Madrid, Spain. EMP acknowledges financial support from UKRI's Future Leaders Fellowship programme [grant MR/V024124/1].

## Author contributions

J.C.M., S.L., E.M.P., and D.G.G. conceived the research. J.C.M. performed the experiments with support from S.G.H. J.C.M. implemented the computational model with support from S.L., E.M.P, and D.G.G. J.C.M., and D.G.G. wrote the original manuscript. J.C.M., S.L., S.G.H., A.A., E.M.P., and D.G.G. conducted the formal analysis and discussion and revised the paper.

## Competing interests

The authors declare no competing interests.
