## [Transparent Peer Review file · Nature Communications]

In-silico platform for the multifunctional design of 3D printed conductive components

Corresponding Author: Professor Daniel Garcia-Gonzalez

Version 0:

Reviewer comments:

Reviewer #1

(Remarks to the Author)

The manuscript reports novel and important insights into the complexity of coupled Multiphysics behaviors of 3D-printed composites with conductive fillers. One of the strength of the study is in the elucidating the governing mechanisms at the relevant scales directly resulting in the overall material performance. The experimental results are accompanied with the numerical simulations providing valuable insights into the driving key mechanisms through rigorous coupled problem formulations. The developed platform effectiveness is then also neatly illustrated by examples of 3D printed with enhanced accuracy thanks to the provided understanding and framework. This study will be of significant interest for the broad scientific community ranging from material fabrication and 3d-printing, to computational and theoretical mechanics societies, as well for soft robotics.

Some technical comments for the consideration of the authors:

- There has been an interest in developing ways to achieve tunable permittivity/conductivity by deformation. For example, the voids were used as the tunable volume fraction “reservoir” (Adv. Mater. Techn. 2200296, 2022) and conductive particle orientation and interactions (IJMS 105880, 2020). The current hints that there is a potential to leverage that through the fabrication control at the small length scale architecture.

-Results in Fig. 1:

-as the deformation is induced, do you add additional charge to maintain the actual field, or is this the reading of potential. That the actual (current electric field) is reported as constant. Please clarify/ updated if needed.

-For the results for the stress field, would that be some thermal stress to be expected to develop in the interval from 0 to 5?

-Some schematics are very small and hard to read – especially, the experimental setup schematics. It could be a good idea to have it also separately and more fully in supporting materials

-Fig.2: is there any physical changes in the interlayer zone, or potential charge accumulation. Is this measurable, or any additional insights that can be gained to inform the modeling?

-Is the additional pre-compression, when the grips (with electrodes are used to connect to the samples). If so, is there any expected/measurable changes in the properties in the area, and potentially affecting the results.

-A minor comment is on the material modeling, in connection to Ref 44 and 45, one could also expect to explore alternatives such micromechanics-based models for transversely isotropic fiber composites, where their convexity / stability is captured through the physical meaningful behavior dictated by the microstructure (as the micro-to-micro is one of the main aspects of the current work).

(Remarks on code availability)

Reviewer #2

(Remarks to the Author)

The paper by Crespo-Miguel et al. presents a multi-scale computational framework for evaluating the thermo-electro-mechanical behavior of printed conductive polymers. The manuscript is well-written, addressing a relevant and timely topic. The results are promising and thoroughly discussed. For these reasons, the reviewer is inclined to support the publication of the paper. However, a final endorsement (and a more thorough assessment) can only be given once the authors have addressed a few critical issues that may impact the consistency of the proposed approach.

Specifically, the reviewer has a major theoretical concern:

- The authors derive parameters from the relationship between the average of the second Piola-Kirchhoff stress and the average of the Green-Lagrange strain (as mentioned on page 11 of the Supplemental Material). However, the relationship between macroscale quantities and homogenized microscale quantities should be rigorously obtained from the Hill-Mandel condition, which can be proven for large strains only by starting from the first Piola-Kirchhoff stress and the deformation gradient. The authors may wish to consult the work by Marc Geers et al., Homogenization Methods and Multiscale Modeling: Nonlinear Problems (Encyclopedia of Computational Mechanics, Second Edition). Therefore, if the reviewer's understanding is correct, the scale transition employed by the authors is flawed. Please provide a justification or revise the results accordingly.

Additional significant concerns are as follows:

- A two-scale homogenization approach requires that the separation of scales between the micro and macro problems is maintained. This does not appear to be the case here. Could the authors provide further clarification on this point?
- The coupling between thermo-mechanics and electro-mechanics is introduced by assuming that model parameters depend on the respective fields, both at the micro-scale and macro-scale. While this approach is legitimate at the micro-scale, the reviewer questions its validity at the macro-scale. A more thorough coupling mechanism may arise, and it should be properly accounted for. In fact, the homogenization process may indicate that macroscale stresses depend explicitly on temperature and electric field. In other words, a fully consistent homogenization procedure should not "postulate" a macroscale response, as the authors seem to do, for instance, in Eq. (12) and subsequent sections. The authors should clarify the theoretical foundation of their approach and ensure it aligns with existing literature. Additionally, they should carefully justify all modeling choices at the macroscale, demonstrating that these are consistent with the numerically obtained microscale responses.
- The application showcased in the "Smart Selector of Printing Parameters to Design Multi-Functional Structural Components" section is both impressive and engaging. However, only a limited number of results are presented, many of which are discussed qualitatively. The impact of the paper would be significantly enhanced if more quantitative results were included.

Lastly, the reviewer would like to highlight the following minor points:

- The line styles used in Fig. 3A-C make the figures difficult to interpret. Additionally, the legends and captions do not clearly explain the meaning of the grey regions. Clarifying this would improve the readability of the figures.
- In Eq. (18), the divergence of the gradient of the temperature field should be present, but it appears that the gradient of the gradient is written instead, based on standard continuum mechanics notation.

(Remarks on code availability)

Reviewer #3

(Remarks to the Author)

Please, refer to the attached pdf file

(Remarks on code availability)

Having been implemented in FEniCS, the code can be linked to the relevant sections (weak forms) of the supplementary material, which I find particularly compelling.

Version 1:

Reviewer comments:

Reviewer #1

(Remarks to the Author)

The authors have carefully revised the manuscript and addressed most of the comments of the reviewers in the revision and/or in their response. The revised manuscript can be recommended for publication in Nature Communication.

As a minor suggestions, which can be addressed during the publication process of the paper, the author may consider providing some user guidelines for the use of the numerical code that the authors make available online (which is very much appreciated).

(Remarks on code availability)

As a minor suggestions, which can be addressed during the publication process of the paper, the author may consider providing some user guidelines for the use of the numerical code that the authors make available online (which is very much appreciated).

Reviewer #2

(Remarks to the Author)

The authors have effectively addressed my previous observations, leading to significant improvements in the manuscript. I believe the paper is now suitable for publication in its current form.

(Remarks on code availability)

Reviewer #3

(Remarks to the Author)

The authors have addressed all my comments and hence, I recommend its publication in NCOMMS in its present form

(Remarks on code availability)

Notes on revision made to manuscript NCOMMS-24-44509

Response to the reviewers

Reviewer 1

The manuscript reports novel and important insights into the complexity of coupled Multiphysics behaviors of 3D-printed composites with conductive fillers. One of the strength of the study is in the elucidating the governing mechanisms at the relevant scales directly resulting in the overall material performance. The experimental results are accompanied with the numerical simulations providing valuable insights into the driving key mechanisms through rigorous coupled problem formulations. The developed platform effectiveness is then also neatly illustrated by examples of 3D printed with enhanced accuracy thanks to the provided understanding and framework. This study will be of significant interest for the broad scientific community ranging from material fabrication and 3d-printing, to computational and theoretical mechanics societies, as well for soft robotics.

We appreciate the positive opinion of the reviewer in his/her evaluation and the constructive feedback to improve the quality of our work.

Some technical comments for the consideration of the authors:

Reviewer Comment 1.1 — There has been an interest in developing ways to achieve tunable permittivity/conductivity by deformation. For example, the voids were used as the tunable volume fraction “reservoir” (Adv. Mater. Techn. 2200296, 2022) and conductive particle orientation and interactions (IJMS 105880, 2020). The current hints that there is a potential to leverage that through the fabrication control at the small length scale architecture.

Reply: We thank the reviewer for his/her comment. The study of the effect of the printing process in the electrical permittivity of this type of materials would be, definitely, of great interest as future work. The recommended articles have been cited in the introduction section.

Reviewer Comment 1.2 — Results in Fig.1. As the deformation is induced, do you add additional charge to maintain the actual field, or is this the reading of potential. That the actual (current electric field) is reported as constant. Please clarify/ updated if needed.

Reply: We thank the reviewer for highlighting this. The effective resistivity (ρ_{eff}) is calculated in terms of the electric potential drop measured throughout the experiment. This voltage drop (V_{drop}) is measured in an additional in-series resistor (R_{ref}) while applying a constant reference voltage (V_o).

$$\rho_{eff} = \frac{R_i A}{L} \quad ; \quad R_i = \frac{R_{ref}(V_o - V_{drop})}{V_{drop}} \quad (1)$$

Thus, the effective resistivity and electric field are calculated in the reference configuration, as the reference voltage does not vary. We would like to highlight that the computational values are also displayed in the reference configuration, to correctly compare experimental and numerical results. This has been clarified in the figure caption:

Figure captions 1 and 3: “To provide fair comparisons between experimental and modelling data, as the voltage at reference configuration is kept constant during the tests, the effective resistivity and electric field are calculated in the reference configuration too.”

Reviewer Comment 1.3 — Results in Fig.1. For the results for the stress field, would that be some thermal stress to be expected to develop in the interval from 0 to 5?

Reply: We thank the reviewer for this pertinent comment. As suggested by the reviewer, this constrain introduces internal thermal stress due to electro-thermal heating. Nevertheless, those stresses were negligible, being lower than 1% of the peak stress. Due to the scale used in the figure, this contribution is not clearly observed (we have introduced a small offset in the Y-axis to allow for this visualization).

Furthermore, for consistency with the simulations, in the fully-coupled tests we replicated these boundary conditions (BCs) at the macroscale, imposing these BCs to impede deformations along the axial sample direction.

This point has been clarified in Results section: "The boundary conditions during this stage impede the sample expansion in the axial direction due to electro-thermal heating. This introduces thermal stresses within the sample but these are negligible, being lower than 1% of the peak stress."

Reviewer Comment 1.4 — Results in Fig.1. Some schematics are very small and hard to read – especially, the experimental setup schematics. It could be a good idea to have it also separately and more fully in supporting materials

Reply: We appreciate the reviewer's suggestion. An additional subsection has been added to Supplemental Information, explaining the experimental setup for thermo-electro-mechanical experiments and adding a general schematic:

Figure S.9 Complete experimental set-up diagram for thermo-electro-mechanical tests. 1) Universal testing machine; 2) Infrared camera; 3) PLA/CB filament sample; 4) DC power source; 5) Known resistor; 6) DAQ system; 7) Computer, recording voltage data from DAQ system; 8) Signal trigger to control the electrical and mechanical recording.

Reviewer Comment 1.5 — Results in Fig.2. Is there any physical changes in the interlayer zone, or potential charge accumulation? Is this measurable, or any additional insights that can be gained to inform the modeling?

Reply: We thank the reviewer for his/her question. Please note that the identification and quantification of multifunctional changes in these regions is one of the contributions of the modelling framework to this work. In the literature, previous work has shown alterations in the degree of crystallinity with respect to the filament region [1, 2]. These are due to the imperfect polymeric chains diffusion within adjacent filament regions during their deposition [3]. These changes impact the distribution of conductive particles within these regions and, considering the role of such particle distribution modulating the tunnelling effect [4], an impact on conductive properties is expected. However, to the best of our knowledge, there are no experimental techniques available to characterise the electro-mechanical behaviour of these materials at that scales (some approaches could be modified to evaluate static properties but these would not enable measurements under stretching conditions).

Thanks to the proposed two-scale optimisation method, we demonstrated and quantified the degradation in conductive properties showing that this consideration is needed to reproduce the experimental results.

We have added this discussion in Results section: "Note that this methodology is not based on bottom-up homogenisation approaches [5–7], but the mesoscopic homogenisation is used to get insights that are unfeasible to capture with current experimental methods. This strategy allows to overcome the experimental difficulties needed to identify potential degradation in the multifunctional properties within the filament-to-filament adhesion regions and quantify them by direct comparison with macroscopic experimental data."

Reviewer Comment 1.6 — Is the additional pre-compression, when the grips (with electrodes are used to connect to the samples). If so, is there any expected/measurable changes in the properties in the area, and potentially effecting the results.

Reply: We appreciate the reviewer comment. As suggested by the reviewer, a transverse pre-compression at the gripping regions is needed to ensure proper electrical contact between the sample and the electrodes. We experimentally defined this gripping condition by the minimum pre-compression enabling stable electric conduction and avoiding sample sliding during the tensile test. We further evaluated potential excessive stress concentration at the gripping regions by confirming that the samples break occurs far from these regions.

The stable condition for electric conduction was defined as the pre-compression leading to negligible resistivity changes for larger pre-compression values. This response can also be observed from the results shown in Fig. S7. Here, we performed a mechano-electrical test under uniaxial compression. The results show that proper electrical contact is not achieved until reaching a trigger deformation ($\approx 2 - 5\%$).

Figure S7. Mechano-electrical compressive tests of PLA/CB samples. Uniaxial compressive tests were carried out while measuring the electrical current flowing through the samples. The solid/dashed lines represent the average values of the response, whereas the shaded areas represent the experimental deviation. Three samples were considered for each experimental condition. The directions considered for the tests are: longitudinal, transverse and vertical.

This has been clarified in the manuscript, in Methods: "A pre-compression at the gripping regions was applied to ensure proper electrical contact between the sample and the electrodes. We experimentally defined this gripping condition by the minimum pre-compression enabling stable electric conduction and avoiding sample sliding during the tensile test. We further evaluated potential excessive stress concentration at the gripping regions by confirming that the samples break occurs far from these regions."

In addition, the vast majority of electric current in the grips will be localized in the zones nearer to the free boundary (see Fig. R1). This is caused by the "infinite" conductivity of the grippers' conductive plates, which close the electrical circuit in the first points of contact with the sample. In this regard, the changes in resistivities in the zones attached to the grips will not substantially modify the mechano-electrical behaviour under uniaxial tension.

Figure R1. Diagram of the electrical contact and localisation of the current densities

Reviewer Comment 1.7 — A minor comment is on the material modeling, in connection to Ref 44 and 45, one could also expect to explore alternatives such micromechanics-based models for transversely isotropic fiber composites, where their convexity / stability is captured through the physical meaningful

behavior dictated by the microstructure (as the micro-to-macro is one of the main aspects of the current work).

Reply: We thank the reviewer for his/her comment. At the beginning of the work the use of transversely isotropic models was contemplated. Nonetheless, experimental evidence pointed out that both mechanical and electrical properties were different in the three principal directions of the printed samples, i.e., along the filament, in between adjacent filaments (inter-filament) and in between subsequent layers (inter-layer). This is explained by the different thermal conditions during printing between filaments in the same layer (higher effective temperature at inter-filament) and filaments in different layers (inter-layer). In this regard, the use of transversely isotropic models lacks from the variation in properties in one of those directions. Therefore, we considered that, for our continuum model, the use of an orthotropic approach was essential.

Reviewer 2

The paper by Crespo-Miguel et al. presents a multi-scale computational framework for evaluating the thermo-electro-mechanical behavior of printed conductive polymers. The manuscript is well-written, addressing a relevant and timely topic. The results are promising and thoroughly discussed. For these reasons, the reviewer is inclined to support the publication of the paper. However, a final endorsement (and a more thorough assessment) can only be given once the authors have addressed a few critical issues that may impact the consistency of the proposed approach.

We appreciate the positive opinion of the reviewer in his/her evaluation and the constructive feedback to improve the quality of our work.

Specifically, the reviewer has a major theoretical concern:

Reviewer Comment 2.1 — The authors derive parameters from the relationship between the average of the second Piola-Kirchhoff stress and the average of the Green-Lagrange strain (as mentioned on page 11 of the Supplementary Information). However, the relationship between macroscale quantities and homogenized microscale quantities should be rigorously obtained from the Hill-Mandel condition, which can be proven for large strains only by starting from the first Piola-Kirchhoff stress and the deformation gradient. The authors may wish to consult the work by Marc Geers et al., Homogenization Methods and Multiscale Modeling: Nonlinear Problems (Encyclopedia of Computational Mechanics, Second Edition). Therefore, if the reviewer’s understanding is correct, the scale transition employed by the authors is flawed. Please provide a justification or revise the results accordingly.

Reply: We thank the reviewer for highlighting such an important point for providing a robust and rigorous framework. We agree with the reviewer and we have modified the analyses accordingly. To this end, we have now computed the macroscopic quantities of the first Piola-Kirchhoff stress and deformation gradient by averaging their corresponding local quantities following the Hill-Mandel condition or macrohomogeneity condition [8]. These macroscopic homogenized values have then been used to compute the second Piola-Kirchhoff stress and Green-Lagrange strain tensors at the macroscale. These latter macroscopic tensors have finally been used to calculate the corresponding shear and Young’s moduli and Poisson’s ratios. The results have been recomputed and no significant changes in the results have been obtained, with variations of less than 1e-6 % (due to the small range of deformations/rotations and the symmetries contained in the mesoscale model). However, we agree with reviewer that the formulation and method is much more robust and rigorous now.

To amend the document, this modification has been added to the manuscript. These changes can be found in the updated section: “Supplementary information. Identification of macroscopic elastic parameters via homogenization”, and we have introduced this discussion in the main text.

“To obtain the Young’s moduli and Poisson’s ratios, uniaxial tensile simulations were performed at the mesoscale to calculate each parameter based on the average of the desired solution fields. E_i is calculated as the slope between the macroscopic Second Piola-Kirchhoff stress at the i^{th} direction ($\overline{\mathbf{S}}_{ii}$) and the macroscopic Green-Lagrange strain at the i^{th} direction ($\overline{\mathbf{E}}_{ii}$). To be consistent with the Hill-Mandel or macro-homogeneity condition [8, 9], $\overline{\mathbf{S}}_{ii}$ and $\overline{\mathbf{E}}_{ii}$ were computed from the averaged deformation gradient ($\overline{\mathbf{F}}_{ij} = \frac{1}{\Omega} \int_{\Omega_o} \mathbf{F}_{ij} d\Omega$) and the first Piola-Kirchhoff stress ($\overline{\mathbf{P}}_{ij} = \frac{1}{\Omega} \int_{\Omega_o} \mathbf{P}_{ij} d\Omega$). ν_{ji} is calculated as the slope between the average of Green-Lagrange strain in the j^{th} direction with respect to the i^{th} one, where the uniaxial loading is applied. In the case of the shear moduli obtention, pure shear conditions on each principal plane were applied to RVEs. Then, they were calculated as the slope between the average of the ij^{th} component of the Second Piola-Kirchhoff stress tensor with respect to the desired tangential component of the average of Green-Lagrange strain tensor.”

Additional significant concerns are as follows:

Reviewer Comment 2.2 — A two-scale homogenization approach requires that the separation of scales between the micro and macro problems is maintained. This does not appear to be the case here. Could the authors provide further clarification on this point?

Reply: We thank the reviewer for his/her comments. As suggested by the reviewer, our approach does not uniquely make use of the mesoscale homogenization to calibrate the model parameters at both scales. We acknowledge this limitation, and we would like to highlight that our main objective was not to propose a bottom-up method based on the mesoscopic homogenization.

Our main aim is to provide a macroscopic computational framework that can help guiding the multifunctional design of these conductive components and, helped by such model development, provide new insights into the thermo-electro-mechanical response of 3D printing conductive thermoplastics. The problem at hand is very complex as the mechanical, electrical and thermal problems are highly coupled. Moreover, the printing strategy affects some mesostructural features of the components that strongly modulate these couplings.

From an experimental point of view, we are able to characterize some thermo-mechanical responses at the macroscale but, the mechano-electrical and electro-thermal problems are highly influenced by the boundary conditions, making the identification of material parameters, such as effective conductivity, impossible without computational help. In addition, the experimental analysis of the links between such responses and the mesostructural characteristics derived from the printing process is very limited. The importance of the mesoscopic model in this work relies on the capability of serving as virtual testbed for evaluating these missed links and guide the calibration of the model at the macroscale. Please note that, due to the high complexity of the problem, the present combination of methodologies was found to be the best solution to provide a robust and efficient computational platform to guide the design of multifunctional structures.

Taking this into consideration, we have amended the manuscript by indicating this limitation. The section "*In-silico* platform to connect multifunctional responses to printing parameters across scales" now reads:

"The homogenised formulation describes the mechano-electrical problem considering different responses for each phase (Fig. 2C.1). This framework allows for capturing the mechanical and electrical responses depending on the printing parameters and provides homogenised material parameters to feed the macroscopic continuum formulation. The macroscopic formulation accounts for: (i) a temperature dependent orthotropic elasto-viscoplastic response; (ii) a deformation dependent orthotropic electrical conductivity; (iii) Joule heating; and (iv) a transient thermal response considering convective terms. The macroscopic continuum model provides great flexibility in terms of BCs application, allowing for simulating all possible experimental conditions (Fig. 2C.2). Note that this methodology is not based on bottom-up homogenisation approaches [5–7], but the mesoscopic homogenisation is used to get insights that are unfeasible to capture with current experimental methods. This strategy allows to overcome the experimental difficulties needed to identify potential degradation in the multifunctional properties within the filament-to-filament adhesion regions and quantify them by direct comparison with macroscopic experimental data."

Reviewer Comment 2.3 — The coupling between thermo-mechanics and electro-mechanics is introduced by assuming that model parameters depend on the respective fields, both at the micro-scale and macro-scale. While this approach is legitimate at the micro-scale, the reviewer questions its validity at the macro-scale. A more thorough coupling mechanism may arise, and it should be properly accounted for. In fact, the homogenization process may indicate that macroscale stresses depend explicitly on temperature and electric field. In other words, a fully consistent homogenization procedure should not "postulate" a macroscale response, as the authors seem to do, for instance, in Eq. (12) and subsequent sections. The authors should clarify the theoretical foundation of their approach and ensure it aligns with existing literature. Additionally, they should carefully justify all modelling choices at the macroscale, demonstrating that these are consistent with the numerically obtained microscale responses.

Reply: As suggested by the reviewer, the continuum macroscopic framework is not explicitly extracted from the homogenization framework. Note that the mesoscale model here is used to complement the experimental measurements, allowing us to identify the links between the physical couplings and the mesostructured derived from the printing process (i.e., filament orientations, resulting porosity, etc.). The current methodology should be interpreted as a hybrid experimental-computational platform to first characterize the multifunctional behavior of these materials and then provide an efficient designing tool for functional structural components.

We would like to highlight all the complexities of the problem that even go beyond the mesoscale. In this regard, there are some mechanisms such as electrical conductivity through the material that are the result of both meso- and microstructural effects (e.g., tunneling effect at the microscale). Therefore, as necessarily done at the mesoscale to incorporate some assumptions from the microscale, we adopt some assumptions when performing the homogenisation from the mesoscale. Thus, based on previous modelling works [4], mesoscale modelling coupled response and experimental observations, we defined some physically-based, but phenomenological, dependences at the macroscale to simplify the final modelling framework and facilitate the links with

the printing parameters choice. This was essential to allow effective and fast optimizations at the functional and structural levels.

Nonetheless, it is important to remark that the mechano-electrical parameters were extracted from the homogenisation framework (i.e., independent of the macroscopic complex BCs). These have been used to feed the macroscopic continuum framework and provide a direct relationship with printing parameters, overcoming the experimental limitations to evaluate these mesostructural properties.

Reviewer Comment 2.4 — The application showcased in the "Smart Selector of Printing Parameters to Design Multi-Functional Structural Components" section is both impressive and engaging. However, only a limited number of results are presented, many of which are discussed qualitatively. The impact of the paper would be significantly enhanced if more quantitative results were included.

Reply: We thank the reviewer for his/her appreciation of the work. Following this suggestion, we have incorporated relevant quantitative data in Fig.5.

New Figure 5:

Figure 5. Application of the in-silico platform to the optimal design of a 3D printed heatable cartridge for DIW printers. **a** Design of a 3D printed heatable cartridge used to enclose a conventional syringe containing a magnetorheological ink for DIW printers. Controlled increases in temperature improve the printability of the ink enhancing its polymerisation. **b** Temperature distribution measured after the application of 30V for 10 min, showing numerical and experimental results for two printing strategies. **c** Optimised solution provided by the *in-silico* platform in terms of local printing directions to provide a uniform distribution of temperatures. The inclination angle with respect to the printing bed is defined by α_x , while the proper printing direction of each cartridge division (i) is defined by α_z^i . **d** Comparison of the temperature profile obtained experimentally and numerically from the optimised heatable cartridge. The average surface temperature is represented in the graph along with its standard deviation. **e** Comparison of the experimental printing resolution improvement when using the heating cartridge. **f** Printing of a magnetoactive gripper. This gripper closes when a magnetic field is applied.

Additionally, we have included a new section and associated figure in the Supplementary Information to better exemplify and explain this point.

New section in Supplementary Information:

Heatable cartridge, thermal images and model data post-processing

"This subsection provides an overview of the numerical and experimental post-processing of the temperature data taken from the heatable cartridge. As defined in the main manuscript, section "Smart selector of printing parameters to design multi-functional structural components", an optimal set of printing directions was sought to minimize the heating heterogeneity of the component. To this end, the standard deviation of the nodal temperatures of the cartridge was defined as the objective function to minimize. In this case, every node in the mesh was considered to calculate the temperature standard deviation. Nonetheless, to correctly compare the numerical results with the thermal IR images, a different set of nodes was selected. As shown in Fig. S10.B, the IR images are taken from one side of the heating cartridge. Therefore, we only used the side surface nodes to compute the average and the standard deviation of the temperatures during the heating process."

A) Nodal selection for printing optimisation

B) Nodal selection for experimental/numerical comparison

Figure S10. A) Nodal selection of the whole mesh to optimise the heatable cartridge printing. B) Nodal selection of the side visible nodes, to correctly compare the IR image data with the numerical solution.

Lastly, the reviewer would like to highlight the following minor points:

Reviewer Comment 2.5 — The line styles used in Fig. 3A-C make the figures difficult to interpret. Additionally, the legends and captions do not clearly explain the meaning of the grey regions. Clarifying this would improve the readability of the figures.

Reply: We have modified the line styles used in the mentioned figure, as well as the shaded areas. In addition, we have added references to the experimental deviation, shown by the grey regions, both in the legend and the figure caption. The modified figure and caption are presented as follows:

Figure 3. Predictive capabilities of the multi-physical in-silico platform. **a** Simulations of thermo-mechanical tests at a temperature of 45°C on longitudinal, transverse and oblique samples. **b** Simulations of mechano-electrical tests on longitudinal, transverse and oblique samples. **c** Simulations of electro-thermal tests applying an electric field (\mathbb{E}) of 250 V/m on longitudinal, transverse and oblique samples. **d** Simulation of a thermo-electro-mechanical test applying an electric field (\mathbb{E}) of 187.5 V/m and uniaxial tensile loading on a longitudinal sample. Experimental (black) and numerical (blue/green/red depending on the printing orientation) results are presented together in all graphs. The grey shaded areas represent the experimental deviation of the tests. To provide fair comparisons between experimental and modelling data, as the voltage at reference configuration is kept constant during the tests, the effective resistivity and electric field are calculated in the reference configuration too.

Reviewer Comment 2.6 — In Eq. (18), the divergence of the gradient of the temperature field should be present, but it appears that the gradient of the gradient is written instead, based on standard continuum mechanics notation.

Reply: We thank the reviewer for pointing this typo out. We have amended the document by modifying the expression as follows: Manuscript Eq(18) and Supplementary Eq(55):

$$\rho\dot{\theta} - \nabla_{\mathbf{x}} \cdot (\kappa J_F \mathbf{F}^{-1} \cdot \mathbf{F}^{-T} \cdot \nabla_{\mathbf{x}} \theta) - \mathbb{J} \cdot \mathbb{E} + h(\theta - \theta^{\text{ref}}) = 0 . \quad (2)$$

Reviewer 3

The paper In-silico tool for the multifunctional design of 3D printed conductive components presents a robust multi-scale computational framework for assessing the coupled thermo-electro-mechanical behaviour of printed conductive polymers. It is well-written, drawing upon a diverse range of expertise including continuum mechanics, computational modelling, and experimental analysis and provides a thorough, insightful interpretation of both computational and experimental results. Given the paper's substantial contribution to a highly relevant topic, I believe it should be considered for publication in NCCOMS, contingent upon the authors addressing a few minor modifications and suggestions.

We appreciate the positive opinion of the reviewer in his/her evaluation and the constructive feedback to improve the quality of our work.

Reviewer Comment 3.1 — While the paragraphs are well-written, some of them feel somewhat lengthy. I recommend that the authors consider shortening certain paragraphs to enhance readability.

Reply: We thank the reviewer for his/her assessment. We agree with the reviewer. The document has been amended by shortening certain paragraphs of the manuscript.

Reviewer Comment 3.2 — In Figure 1 (and in the remaining figures), the caption references certain variables—such as electric field, stretch, and effective resistivity—using their descriptive names. However, in the plots, these quantities are represented symbolically as E , λ and ρ_{eff} . To improve clarity and facilitate interpretation of the figures and captions, I suggest that each variable mentioned in the caption be accompanied by its mathematical symbol. For example, “electric field (E)”.

Reply: We thank the reviewer for highlighting this point. We agree with the reviewer and the document has been amended accordingly.

Reviewer Comment 3.3 — In the supplementary material, around equation (2), the term “deviatoric” is used. While this term is commonly understood among researchers in certain fields, the multidisciplinary audience of this work may include readers unfamiliar with its meaning. A brief definition of “deviatoric” would enhance clarity for all readers.

Reply: We appreciate the reviewer's comment. The document has been amended, including a brief definition of the “deviatoric” stress, reading as:

“Where σ_{dev} denotes the deviatoric part of the Cauchy stress tensor, defined as the difference between the total stress and its hydrostatic part ($\sigma_{dev} = \sigma - (tr(\sigma)/3)\mathbf{1}$).”

Reviewer Comment 3.4 — What is $\dot{\epsilon}_o$ in equation (7) in the supplementary material. Although it is later calibrated in Table S3, this parameter is not explicitly referenced or defined in the text immediately following the equation.

Reply: We appreciate the reviewer's comment. The term $\dot{\epsilon}_o$ is defined as the reference strain rate from which the strain rate sensitivity is then calibrated. Uniaxial tensile tests at three different strain rates were performed on PLA/CB filament samples (10^{-4} , 10^{-3} and 10^{-2} s^{-1}). The minimum of these values was taken as the reference strain rate, $\dot{\epsilon}_o$.

This clarification has been added to Supplementary information, reading as follows:

“with C being a model parameter describing the strain rate dependency and $\dot{\epsilon}_o$ the reference strain rate. The former parameter is calibrated using experimental data obtained from uniaxial tensile tests performed to PLA/CB filament samples at three different strain rates (10^{-4} , 10^{-3} and 10^{-2} s^{-1}). The latter parameter is defined as the lowest strain rate used in those experiments (10^{-4} s^{-1}).”

Reviewer Comment 3.5 — Immediately before equation (12) in the supplementary material, the authors state “...with the first invariant of deformation gradient, I_1^F (see):”. It appears that a reference is missing within the parentheses.

Reply: We thank the reviewer for highlighting this typo. The missing reference has been added to Supplementary Information.

Reviewer Comment 3.6 — In equation (29) of the supplementary material, the authors propose an approach where the plastic strain \mathbf{F}_p is introduced as an additional unknown in the system of equations, thereby requiring spatial discretization. However, alternative methodologies exist in which \mathbf{F}_p is treated as an internal variable, allowing it to be solved independently at each Gauss point within the domain. The authors may wish to briefly comment on this alternative approach, as well as their rationale for choosing the current method, which is particularly convenient for implementation within the FEniCS framework.

Reply: We appreciate the reviewer comment. Even though several authors have treated F_p as an internal variable when using different FEA software, this approach is not that straightforward when using FEniCS. We devised a solution in previous works by considering the \mathbf{F}_p components as extra degrees of freedom and including the viscoplastic flow rule as an additional residual [4, 10, 11]. While this solution is not optimal in terms of computational cost, it offers a highly flexible framework for automatic numerical differentiation. This flexibility enables effortless testing of various constitutive models without the need to derive stiffness matrices manually. This is explained in the Supplementary Information.

Reviewer Comment 3.7 — The authors may wish to discuss their choice of the anisotropic model in equation (30) of the supplementary material, which extends the Saint-Venant model to anisotropic conditions. Specifically, it would be beneficial to address why this model was chosen over finite strain models that avoid the known limitations of the Saint-Venant model in the finite strain regime.

Reply: We thank the reviewer for his/her comment. After an extensive literature review, a general set of polyconvex free energy functions for hyperelastic orthotropic materials was found [12]. In this regard, to fulfil the polyconvexity conditions ($\psi = \hat{\psi}(\mathbf{F}, \text{adj}\mathbf{F}, J)$) this set of energy functions depends on modified invariants,

$$I_1^i = \text{tr}(\mathbf{C} \cdot \mathbf{L}_{ii}); \quad I_2^i = \text{tr}[(\text{cof}\mathbf{C}) \cdot \mathbf{L}_{ii}]; \quad I_3 = \det(\mathbf{F}). \quad (3)$$

Nonetheless, we found some issues with this formulation. When using the first and third invariant with the polyconvex energy function, the changes in deformations in each principal direction when applying a pure hydrostatic load cannot be captured. This drawback arises from the use of J in the orthotropic formulation without weighting each principal direction contribution with an "oriented" parameter. Moreover, when employing the Saint-Venant model, the framework is capable of capturing this principal direction dependence by the use of Lamé parameters in each principal direction.

Due to the absence of high compressive stresses, we found Saint-Venant Kirchhoff model ideal for this work. We did not encounter convergence or any other numerical issue in the simulations performed herein.

References

1. Collinson, D. W., von Windheim, N., Gall, K. & Brinson, L. C. Direct evidence of interfacial crystallization preventing weld formation during fused filament fabrication of poly(ether ether ketone). *Addit. Manuf.* **51**, 102604. ISSN: 2214-8604 (2022).
2. Pu, J., McIlroy, C., Jones, A. & Ashcroft, I. Understanding mechanical properties in fused filament fabrication of polyether ether ketone. *Addit. Manuf.* **37**, 101673 (2021).
3. Das, A., Gilmer, E. L., Biria, S. & Bortner, M. J. Importance of Polymer Rheology on Material Extrusion Additive Manufacturing: Correlating Process Physics to Print Properties. *ACS Appl. Polym. Mater.* **3**, 1218–1249 (2021).
4. Crespo-Miguel, J., Lucarini, S., Arias, A. & Garcia-Gonzalez, D. Thermo-electro-mechanical microstructural interdependences in conductive thermoplastics. *npj Comput Mater* **9**, 134 (2023).
5. Sozio, F., Lallet, F., Perriot, A. & Lopez-Pamies, O. The nonlinear elastic response of bicontinuous rubber blends. *Int J Solids Struct* **290**, 112660 (2024).
6. Lefèvre, V., Danas, K. & Lopez-Pamies, O. A general result for the magnetoelastic response of isotropic suspensions of iron and ferrofluid particles in rubber, with applications to spherical and cylindrical specimens. *J. Mech. Phys. Solids* **107**, 343–364 (2017).
7. Mukherjee, D., Bodelot, L. & Danas, K. Microstructurally-guided explicit continuum models for isotropic magnetorheological elastomers with iron particles. *Int. J. Non-Linear Mech.* **120**, 103380 (2020).
8. Hill, R. Elastic properties of reinforced solids: Some theoretical principles. *J. Mech. Phys. Solids* **11**, 357–372 (1963).
9. Geers, M. G. D., Kouznetsova, V. G., Matouš, K. & Yvonnet, J. in *Encyclopedia of Computational Mechanics Second Edition* 1–34 (John Wiley Sons, Ltd). ISBN: 9781119176817.
10. Magneto-diffusion-viscohyperelasticity for magneto-active hydrogels: Rate dependences across time scales. *J. Mech. Phys. Solids* **139**, 103934 (2020).
11. Lucarini, S., Moreno-Mateos, M., Danas, K. & Garcia-Gonzalez, D. Insights into the viscohyperelastic response of soft magnetorheological elastomers: Competition of macrostructural versus microstructural players. *Int J Solids Struct* **256**, 111981 (2022).
12. Itskov, M. & Aksel, N. A class of orthotropic and transversely isotropic hyperelastic constitutive models based on a polyconvex strain energy function. *Int. J. Solids Struct.* **41**, 3833–3848. ISSN: 0020-7683 (2004).

Notes on reviewers response to manuscript NCOMMS-24-44509

Response to the reviewers

Reviewer 1

The authors have carefully revised the manuscript and addressed most of the comments of the reviewers in the revision and/or in their response. The revised manuscript can be recommended for publication in Nature Communication.

As a minor suggestions, which can be addressed during the publication process of the paper, the author may consider providing some user guidelines for the use of the numerical code that the authors make available online (which is very much appreciated).

We appreciate the positive opinion of the reviewer in his/her evaluation and the constructive feedback to improve the quality of our work.

We have included the DOI to the code repository in Code Ocean. In addition, we have commented the code in detail so that it can be easily understood.

Reviewer 2

The authors have effectively addressed my previous observations, leading to significant improvements in the manuscript. I believe the paper is now suitable for publication in its current form.

We thank the reviewer for his/her positive evaluation and the constructive feedback to improve the quality of our work.

Reviewer 3

The authors have addressed all my comments and hence, I recommend its publication in NCOMMS in its present form.

We appreciate the positive opinion of the reviewer in his/her evaluation and the constructive feedback to improve the quality of our work.

Manuscript NCOMMS-24-44509

The paper *In-silico tool for the multifunctional design of 3D printed conductive components* presents a robust multi-scale computational framework for assessing the coupled thermo-electro-mechanical behaviour of printed conductive polymers. It is well-written, drawing upon a diverse range of expertise—including continuum mechanics, computational modelling, and experimental analysis—and provides a thorough, insightful interpretation of both computational and experimental results. Given the paper’s substantial contribution to a highly relevant topic, I believe it should be considered for publication in NCCOMS, contingent upon the authors addressing a few minor modifications and suggestions.

Comments

1. While the paragraphs are well-written, some of them feel somewhat lengthy. I recommend that the authors consider shortening certain paragraphs to enhance readability.
2. In Figure 1 (and in the remaining figures), the caption references certain variables—such as electric field, stretch, and effective resistivity—using their descriptive names. However, in the plots, these quantities are represented symbolically as E , λ and ρ_{eff} . To improve clarity and facilitate interpretation of the figures and captions, I suggest that each variable mentioned in the caption be accompanied by its mathematical symbol. For example, “electric field (E)”.
3. In the supplementary material, around equation (2), the term “deviatoric” is used. While this term is commonly understood among researchers in certain fields, the multidisciplinary audience of this work may include readers unfamiliar with its meaning. A brief definition of “deviatoric” would enhance clarity for all readers.
4. What is ϵ_0 in equation (7) in the supplementary material. Although it is later calibrated in Table S3, this parameter is not explicitly referenced or defined in the text immediately following the equation.
5. Immediately before equation (12) in the supplementary material, the authors state “...with the first invariant of deformation gradient, I_1^F (see):”. It appears that a reference is missing within the parentheses.
6. In equation (29) of the supplementary material, the authors propose an approach where the plastic strain F_p is introduced as an additional unknown in the system of equations, thereby requiring spatial discretization. However, alternative methodologies exist in which F_p is treated as an internal variable, allowing it to be solved independently at each Gauss point within the domain. The authors may wish to briefly comment on this alternative approach, as well as their rationale for choosing the current method, which is particularly convenient for implementation within the FEniCS framework.
7. The authors may wish to discuss their choice of the anisotropic model in equation (30) of the supplementary material, which extends the Saint-Venant model to anisotropic conditions. Specifically, it would be beneficial to address why this model was chosen over finite strain models that avoid the known limitations of the Saint-Venant model in the finite strain regime.